# Exogenous Proline Enhances Systemic Defense against Salt Stress in Celery by Regulating Photosystem, Phenolic Compounds, and Antioxidant System

**DOI:** 10.3390/plants12040928

**Published:** 2023-02-17

**Authors:** Yanqiang Gao, Jing Zhang, Cheng Wang, Kangning Han, Lixia Hu, Tianhang Niu, Yan Yang, Youlin Chang, Jianming Xie

**Affiliations:** College of Horticulture, Gansu Agricultural University, Lanzhou 730070, China

**Keywords:** celery, salt stress, photosystem, phenolic compounds, antioxidant system

## Abstract

This study aimed to explore how exogenous proline induces salinity tolerance in celery. We analyzed the effects of foliar spraying with 0.3 mM proline on celery growth, photosystem, phenolic compounds, and antioxidant system under salt stress (100 mM NaCl), using no salt stress and no proline spraying as control. The results showed that proline-treated plants exhibited a significant increase in plant biomass due to improved growth physiology, supported by gas exchange parameters, chlorophyll fluorescence, and Calvin cycle enzyme activity (Ketosasaccharide-1,5-diphosphate carboxylase and Fructose-1,6-diphosphate aldolase) results. Also, proline spraying significantly suppressed the increase in relative conductivity and malondialdehyde content caused by salt stress, suggesting a reduction in biological membrane damage. Moreover, salt stress resulted in hydrogen peroxide, superoxide anions and 4-coumaric acid accumulation in celery, and their contents were reduced after foliar spraying of proline. Furthermore, proline increased the activity of antioxidant enzymes (superoxide dismutase, peroxidase, and catalase) and the content of non-enzymatic antioxidants (reduced ascorbic acid, glutathione, caffeic acid, chlorogenic acid, total phenolic acids, and total flavonoids). Additionally, proline increased the activity of key enzymes (ascorbate oxidase, ascorbate peroxidase, glutathione reductase, and dehydroascorbate reductase) in the ascorbic acid–glutathione cycle, activating it to counteract salt stress. In summary, exogenous proline promoted celery growth under salt stress, enhanced photosynthesis, increased total phenolic acid and flavonoid contents, and improved antioxidant capacity, thereby improving salt tolerance in celery.

## 1. Introduction

Salt stress is a major environmental stress that restricts the growth and yield of crops worldwide [1]. Soil salinization spreads globally at a rate of 2 million hectares per year and is a major factor in land degradation in arid and semiarid regions [2]. Approximately 7% of the global terrestrial area is currently affected by salt stress [3]. Low salt stress may decrease plant growth, and high salt stress may lead to significant stagnation of plant growth [4]. Moreover, salt stress affects plant growth and productivity at all developmental stages [5]. The most common effects include disrupting the reactive oxygen species (ROS) detoxification system, membrane damage and plasmolysis [6]. Notably, salt-induced oxidative stress in vegetables may affect their quality and yield [7]. Plants have different defense strategies to reduce the toxic effects of salt stress, such as scavenging active ROS by promoting the ASA–GSH cycle and accumulating free proline and soluble phenolics [8].

Spraying exogenous substances is a method for improving salt tolerance and the quality of plants [9]. Proline has various roles in non-stress and stress conditions and is one of the amino acids that make up proteins [10]. Depending on the initial substrate, proline synthesis includes the ornithine and glutamate pathways [11]. However, in plants, proline is mainly synthesized by the glutamate pathway [12]. Considerable research evidence has shown that proline plays an essential role in response to abiotic stress conditions, such as salinity, high temperature, drought, and others, where an accumulation of proline has been observed in the plant when exposed to these adverse conditions [13,14]. Also, a positive correlation has been described between proline accumulation and plant tolerance to various abiotic stresses [15]. Proline can scavenge free radicals, regulate osmosis, and stabilize plant proteins and membranes [16]. Wani et al. [17] found that foliar spraying with a 20 mM proline solution could increase the length, biomass and leaf area of *Brassica juncea*. Messedi et al. [18] reported that proline application in *Cakile maritima* grown under 200 mM NaCl promoted growth, increased photosynthetic activity, polyphenol content, and quantum yield of photosystem II (ΦPSII), and decreased non-photochemical quenching. Shahid et al. showed that exogenous proline significantly promoted growth and photosynthesis in pea plants (*Pisum sativum* L.) under salt stress [19]. Nakhaie et al. found that exogenous proline reduced the damage to the PSII function in Aloe plants caused by salt stress [20]. Borgo et al. [21] showed that proline can contribute to the elimination of oxidative stress in tobacco under cadmium stress. Hossain et al. [22] reported that exposure to 300 mM NaCl in mung bean growth medium with 15 mM proline significantly reduced H_2_O_2_ and malondialdehyde (MDA) contents and increased glutathione content and glutathione reductase activity. Nounjan et al. [23] showed that exogenous application of 10 mM proline on salt-stressed Oryza sativa seedlings increased H_2_O_2_ content and decreased the activities of superoxide dismutase (SOD), peroxidase (POD) and catalase (CAT). Naliwajski et al. [24] showed that exogenous proline could increase antioxidant enzyme activity and promote proline metabolism in cucumber leaves under salt stress. However, the regulatory mechanisms underlying proline accumulation require further study.

Celery (*Apium graveliens* L.) is an annual or perennial herb of the *Umbelliferae* family, widely grown worldwide [25], and is an important medicinal vegetable rich in calcium, phosphorus, iron, carotene, vitamins, and other nutrients with high nutritional value. For example, flavonoids in celery inhibit cardiovascular inflammation [26]. Although the effects of proline on plant salt stress have attracted widespread attention, most studies have focused on seed germination, seedling growth, and gas exchange. However, few studies have reported the physiological responses of celery to exogenous proline under salt stress. In this study, we investigated the effects of proline application on the growth and gas exchange parameters, stomatal morphology, Calvin cycle, chlorophyll fluorescence, membrane damage, phenolic compounds, and antioxidant system of celery under salt stress using the “American Celery.” Additionally, we elucidated the correlation between polyphenolic compounds and the ROS-scavenging capacity of the ASA–GSH cycle. We employed exhaustive methods of studying stress to provide a theoretical and technological basis for cultivating high-quality and high-yield celery under saline conditions.

## 2. Results

### 2.1. Changes in Growth and Photosynthetic Pigment Content under Salt Stress

The effects of exogenous proline on celery growth under salt stress are shown in Figure 1. Compared with the control (CK), proline (P) treatment increased plant height, stem diameter, fresh weight, and dry weight. Additionally, NaCl (N) treatment significantly decreased plant height, stem diameter, fresh weight of shoot, fresh weight of root, dry weight of shoot, and dry weight by 21.48%, 25.54%, 51.58%, 140.00%, 42.14%, and 56.20%, respectively. Furthermore, treatment with proline + NaCl (PN) significantly decreased plant height, stem diameter, and fresh weight of shoots of PN by 8.08%, 12.74%, and 16.00%, respectively. Moreover, PN significantly increased the plant height, stem diameter, and fresh and dry weight of the shoot and root compared with N (Figure 1A–C). As shown in Figure 1D, compared with CK, proline increased chlorophyll a and b and total chlorophyll by 12.41%, 7.16%, and 10.89%, respectively. Conversely, N decreased chlorophyll a and b and total chlorophyll by 17.19%, 18.87%, and 17.68%, respectively. Moreover, PN increased chlorophyll content to different degrees.

### 2.2. Leaf Stomata and Gas Exchange Parameters

Significant differences were observed in the stomatal status among the treatments (Figure 2A). Stomatal morphology under different treatments was observed by scanning electron microscopy. The stomata under salt stress in each treatment underwent different degrees of stomatal closure compared with the stomata under normal growth conditions (Figure 2A). Moreover, compared with CK, the greatest degree of closure was observed with N treatment, and the second highest closure with PN treatment was consistent with the lower stomatal conductance (Gs) and transpiration rate (Tr) with N treatment (Figure 2C,D). Furthermore, compared with CK, proline treatment significantly increased Gs and Tr by 29.96% and 33.33%, respectively; however, Tr and intercellular CO_2_ concentration (Ci) were not significantly different. Additionally, N treatment significantly decreased the net photosynthetic rate (Pn), Gs, and Tr by 19.31%, 28.09%, and 42.38%, respectively (Figure 2B,C), and significantly increased Ci by 21.84%. Notably, Pn, Gs, and Tr levels with PN treatment were higher than those with N treatment but lower than those with CK and proline treatments; an opposite trend was observed for Ci (Figure 2E). Compared with N, PN treatment significantly increased Pn, Gs, and Tr levels by 17.22%, 15.66%, and 42.98%, respectively, and significantly decreased Ci by 14.57%.

### 2.3. Calvin Cycle Enzyme Activity

Figure 3 shows the Calvin cycle enzyme activities [Rubisco (1,5-diphosphate carboxylase), GAPDH (3-Glyceraldehyde-phosphate dehydrogenase), FBP (Fructose-1,6-bisphosphatase), FBA (1,6-diphosphate aldolase), and TK (Trans-ketolase)]. Compared with CK, N treatment significantly decreased Rubisco, GAPDH, FBP, FBA, and TK activities by 36.38%, 57.50%, 55.94%, 76.36%, and 65.11%, respectively. Conversely, compared with N, the PN treatment significantly increased Rubisco, GAPDH, FBP, FBA, and Tk activities by 20.28%, 137.37%, 20.35%, 230.38%, and 202.94%, respectively. Lastly, compared with CK, proline decreased TK activity, which was still significantly higher than that with N treatment. 

### 2.4. Chlorophyll Fluorescence Parameters 

Figure 4 shows the effect of exogenous proline on celery leaves’ chlorophyll fluorescence parameters under salt stress. N treatment significantly decreased the maximum efficiency of PSII photochemistry (Fv/Fm), the actual photochemical efficiency of PSII (Y(II)), the coefficient of photochemical quenching (qP), and relative electron transfer rate (ETR) compared with CK; however, PN treatment alleviated this effect to different degrees (Figure 4B–D,F), and no significant difference in Fv/Fm, Y(II) qP, and ETR was observed with proline treatment. Furthermore, compared with CK, N treatment significantly increased the coefficient of non-photochemical quenching (qN); PN treatment mitigated this effect to different degrees (Figure 4E). Moreover, no significant difference in qN was observed with proline treatment. Lastly, compared with N, PN treatment significantly increased Fv/Fm and ETR.

### 2.5. Analysis of Cell Membrane Damage and Osmotic Substances under Salt Stress

As shown in Figure 5A, compared with CK, the relative conductivity (REC) was increased by 120.70% with N treatment and 56.44% with PN; however, no significant difference was observed with proline. Furthermore, PN treatment significantly decreased REC by 29.11% compared with N. Furthermore, compared with CK, the malondialdehyde (MDA) content of leaves and petioles was increased by 92.34% and 42.87% with N treatment, and 37.79% and 30.46% with PN, respectively; however, no significant difference was observed with proline (Figure 5B). PN significantly decreased MDA by 28.36% and 8.89% in the leaves and petioles, compared with N. Additionally, compared with CK, the soluble protein content of leaves and petioles was significantly increased with proline, N, and PN (Figure 5C). Compared with CK, the soluble sugar content of leaves and petioles increased by 6.69% and 110.86% with N, respectively; however, no significant difference was observed with proline treatment (Figure 5D). Lastly, PN significantly decreased the soluble protein and sugar content of leaves and petioles compared with N.

### 2.6. Effect of Exogenous Proline on Phenolic Compounds and Total Phenolic Content in Celery

The PC1 and PC2 axes of proline-treated celery leaves explained 65.9% (39.1% and 26.8%, respectively) of the total variation under salt stress conditions (Figure 6A), and the PC1 and PC2 axes of celery petioles explained 65.3% (44.1% and 21.2%, respectively) of the total variation (Figure 6B). Compared with CK, the total phenolic acid content of leaves was increased by 12.18% with N and 6.94% with PN (Figure 6C). Moreover, compared with N, the total phenolic acid content of leaves increased by 7.34% with PN. Additionally, compared with CK, the total flavone content increased by 11.72% in leaves and decreased by 16.09% in petioles with N treatment (Figure 6F). Compared with N, the total flavone content of PN decreased by 17.18% in leaves and increased by 11.18% in petioles with PN treatment. The heat map includes data for 15 phenolic compound metabolites identified in celery leaves and petioles (Figure 6D,E). Furthermore, celery leaves in the P-treatment group contained higher levels of benzoic acid, ferulic acid, caffeic acid, cynarin and chlorogenic acid than CK. Similarly, celery leaves in the N treatment group contained higher levels of kaempferol, 4-coumaric acid, quercetin, and rutin than CK. Furthermore, celery petioles in the P treatment group contained higher levels of gallic acid, ferulic acid, benzoic acid, and quercetin than CK; similarly, those in the N treatment group contained higher levels of 4-coumaric and gallic acids than CK. Lastly, the PN group’s celery petioles contained higher levels of caffeic and chlorogenic acids than CK.

### 2.7. ROS Content

Nitro blue tetrazolium (NBT) and Diaminobenzidine (DAB) staining were performed to observe the contents of O^2-^ and H_2_O_2_ in celery leaves. Compared with CK, the highest degree of staining was observed in the N treatment group, followed by the PN group (Figure 7A). Additionally, the contents of O^2-^ and H_2_O_2_ with N treatment were higher than those in CK. Moreover, no significant difference was observed in the O^2-^ and H_2_O_2_ contents with proline treatment; however, PN significantly reduced their content (Figure 7B,C). Lastly, the O^2-^ and H_2_O_2_ contents with PN treatment were significantly lower than those with N.

### 2.8. Effect of Exogenous Proline on the Antioxidant Enzyme Activity of Celery under Salt Stress

Figure 8A–C show celery leaves and petioles’ SOD, POD, and CAT activities under a salt environment, respectively. Compared with CK, P treatment significantly increased the SOD, POD, and CAT activities, excluding the SOD activity in the petioles. The SOD, POD, and CAT activities in the PN group showed similar trends. Compared with CK, N treatment significantly decreased the SOD, POD, and CAT activities, excluding the CAT activity in the leaf. Compared with N, the SOD, POD, and CAT activities were increased in the leaves and petiole of the PN group.

### 2.9. Effect of Exogenous Proline on the Content of Non-Enzymatic Antioxidant Substances in Celery under Salt Stress

Figure 9A–D show the non-enzymatic antioxidant-oxidized substances, including reduced ascorbic acid, reduced ascorbic acid (ASA), glutathione (GSH) and oxidized glutathione (GSSG). Compared with CK, ASA, GSH, and GSH/GSSG contents were significantly increased, and GSSG content was decreased in the P treatment group; however, no significant difference in GSH was observed in the petioles. Conversely, ASA, GSH, and GSH/GSSG contents were significantly decreased, and GSSG content was increased in the N treatment group. Lastly, ASA, GSH content, and GSH/GSSG contents were significantly decreased in the PN group but higher than in N. 

### 2.10. Effect of Exogenous Proline on the Activity of Ascorbic Acid–Glutathione Cycle Related Enzymes in Celery under Salt Stress

Figure 10A–D, and E show the related enzyme activities of ascorbate oxidase (AAO), ascorbate peroxidase (APX), glutathione reductase (GR), monodehydroascorbate reductase (MDHAR) and dehydroascorbate reductase (DHAR) respectively, in celery leaves and petioles under salt environments. AAO, APX, and GR activities were significantly higher in P, N, and PN groups than in CK. Furthermore, MDHAR was reduced in proline, N, and PN groups, except in the leaves of the P treatment group, where it was significantly different. Furthermore, DHAR was elevated with proline, PN, and N treatments. Notably, it was elevated in petioles and reduced in leaves of the N group. Lastly, the AAO, APX, GR, MDHAR, and DHAR activities in the leaves and petioles of the PN group were higher than those in the N group.

### 2.11. Pearson’s Correlation Analysis of Phenolic Compounds and Ascorbic Acid–Glutathione Cycle

Several sets of significant (*p* < 0.05) or highly significant (*p* < 0.01) positive correlations were observed between flavonoids and phenolic acids. Highly significant positive correlations were found between total phenolic and chlorogenic acids in celery leaves (r = 1.00) and petioles (r = 0.96), respectively (Appendix A). Furthermore, highly significant positive correlations were found between total flavones and rutin in celery leaves (r = 0.97) and petioles (r = 0.98), respectively (Appendix A). Additionally, highly significant positive correlations were found between phenolic compounds and antioxidant activities. Furthermore, highly significant positive correlations were found between benzoic acid and SOD in celery leaves (r = 0.97) and petioles (r = 0.98), respectively (Appendix A). Lastly, GSH/GSSG was significantly positively correlated with total flavones (r = 1.00) in the celery petioles (Appendix A), and a highly significant positive correlation was observed between benzoic acid and POD in celery leaves (r = 0.98) (Appendix A).

## 3. Discussion

The effects of salt stress on plants mainly include ion toxicity, osmotic stress, and oxidative stress, which inhibits growth and reduces crop yield [7,27]. Biomass is an important factor reflecting salt stress in plants and a reliable indicator of whether and to what extent exogenous substances alleviate stress [28]. In the present study, we confirmed that NaCl (100 mM) severely inhibited the normal growth of celery (Figure 1A–C). However, proline treatment promoted celery’s growth and biomass accumulation under salt stress, consistent with previous findings [13,19,23]. 

Chlorophyll is the primary pigment for photosynthesis in plants [29]. Compared with the control, salt stress significantly decreased the chlorophyll content of celery leaves (Figure 1D), possibly due to excess Na^+^ in celery leaves [30]. Photosynthesis converts light energy into chemical energy and organic matter, releasing oxygen into the atmosphere for plant growth [31]. A decline in plant growth is often associated with a decrease in photosynthetic capacity under stressful conditions [32]. Furthermore, stomata are gas exchange channels between the plant interior and atmosphere that control CO_2_ entry into the leaves for photosynthesis [33]. The reduction in photosynthesis under saline conditions is caused by a combination of stomatal closure and non-stomatal limitations [34]. We found that non-stomatal factors principally caused the salt induced reduction in leaf Pn (Figure 2B) because the salt stress response elevated Ci (Figure 2E), even though Gs decreased (Figure 2C). Additionally, the exogenous application of proline counteracted the negative effects of salt stress on Pn, Gs, and Ci levels in celery leaves (Figure 2B,C,E). Moreover, we confirmed that salt stress closed the stomata of apparent celery leaves, and exogenous proline treatment alleviated the closure (Figure 2A). These results suggest that proline may alleviate salt-induced photosynthetic inhibition primarily by regulating non-stomatal limiting factors, such as reduced photosynthetic pigment content, disrupted chloroplast ultrastructure, reduced photochemical reaction activity, or reduced enzyme activity involved in carbon assimilation processes.

The Calvin cycle is the most important carbon fixation pathway in the biosphere. The Calvin cycle has three stages: the carboxylation of ribulose 1,5-bisphosphate, reduction of 3-phosphoglyceric acid, and regeneration of ribulose 1,5-bisphosphate [35]. Rubisco plays a direct role in photosynthesis, converting free CO_2_ in the atmosphere into energy-storing molecules, such as sucrose [30]. Several studies have shown that limiting the content or activity of Rubisco is a main factor for downregulating photosynthesis under stress [36]. In this study, salt stress significantly reduced Rubisco activity. However, exogenous proline increased the activity; nevertheless, it was lower than that of the CK group (Figure 3A). Furthermore, FBA regulates CO_2_ fixation through the Calvin cycle in chloroplasts and participates in glycolysis and gluconeogenesis in the cytoplasm [37]. The Calvin cycle reversibly catalyzes the conversion of glycerol 3-phosphate and dihydroxy-acetone phosphate to fructose 1,6-diphosphate [38]. In our study, salt stress significantly reduced FBPase and FBA activities; however, exogenous proline increased these activities (Figure 3C,D). Moreover, compared with CK, salt stress significantly decreased GAPDH activity; however, exogenous proline significantly increased this activity (Figure 3B). Additionally, exogenous proline and salt stress significantly decreased the activity of TK compared with CK (Figure 3E). Our results suggest that proline accelerates the carbon assimilation pathway by increasing the photosynthetic efficiency in celery, which may contribute to salt tolerance. This result is consistent with that of Zhong et al. [37]; they found TGase positively regulated photosynthesis by activating the Calvin cycle enzymes and inducing changes in cellular redox homeostasis in tomato.

Chlorophyll fluorescence is often used to measure the effect of stress on photosynthesis [20]. Salt stress significantly reduced the chlorophyll fluorescence parameters, Fv/Fm, Y(II), and qP in celery plants compared with CK (Figure 4B–D). Salt stress may be affected by chronic and dynamic photoinhibition and inhibits PSII in celery plants [39]. Hence, the rapid decrease in Fv/Fm may cause chronic photosuppression (Figure 4B), and the increase in qN may result from dynamic photoinhibition (Figure 4E). Proline applica-tion enhanced fluorescence due to photosynthesis and heat dissipation (Figure 4D) and increased the relative electron transfer rate—an imbalance between light energy uptake and metabolic depletion caused to some extent by salt stress. Additionally, proline reduced the excitation stress faced by PSII (Figure 4F) and inhibited the reduction in Y(II) (Figure 4C). This result is similar to that of Nakhaie et al. [20]. In addition, Messedi et al. [18] showed that under heat stress, the beneficial effect of exogenous proline on the response of *Cakile maritima* to salinity is due to its role in protecting chloroplast structures. Therefore, we assumed that celery’s chloroplast structure under salt stress was protected by the accumulation of proline or its metabolites, which improves the efficiency of light energy uptake, distribution, and utilization by celery plants. 

Changes in MDA content reflect lipid peroxidation levels in cell membranes [40]. Salt stress significantly increased the MDA content; however, proline inhibited its sustained increase (Figure 5B). The relative conductivity showed a similar trend under salt stress (Figure 5A). Therefore, we hypothesized that proline protects the structural integrity of plant cell membranes, which is supported by a similar study on pepper [41]. The simultaneous increase in MDA and relative conductivity indicate an increase in lipid peroxidation of the cell membrane and severe cellular damage. To survive in unfavorable environments, plants adopt adaptive defensive mechanisms, such as osmotic pressure regulation or osmoprotectants [42]. Regulation of osmotic substance accumulation improves plant resistance. Soluble proteins and sugars are important osmotic substances, and their levels are responsive to oxidative damage [43]. We found that soluble protein and sugar contents in celery leaves and petioles were increased under salt stress, which was inhibited by proline; however, their levels were still higher than those in the CK group (Figure 5C,D), consistent with previous studies [43,44]. Therefore, it is essential to use exogenous proline to alleviate the damage caused by salt stress in celery plants. 

Phenolic compounds are an important class of secondary metabolites that play key physiological roles throughout the life cycle of plants [45]. Under abiotic stress conditions, plants synthesize increased phenolic compounds, contributing to their antioxidant capacity [46]. Phenolic compounds include monophenols, polyphenols, oligophenols, and simple phenolic compounds [47]. Xie et al. [46] found that phenolic acids and isoflavones in Brussels sprouts increased by 10.91% and 19.14% under NaCl stress. Kusvuran et al. [27] found that salt stress increased the contents of total flavonoids and phenolic compounds in pepper. We found that exogenous proline significantly increased the total phenolic acids in celery leaves and petioles and total flavones in celery petioles compared with CK. Additionally, salt stress significantly increased the total flavones in celery leaves (Figure 6C,F). 

Phenolic acids can be divided into hydroxybenzoic (p-hydroxybenzoic acid, gallic acid, and protocatechuic acid.) and hydroxycinnamic acids (coumaric acid, ferulic acid, chlorogenic acid, and caffeic acid) [48]. We performed a hierarchical cluster analysis on the entire dataset to elucidate the effect of exogenous proline on phenolic compound metabolites in celery under salt stress [49]. The heat map includes data for 15 phenolic compound metabolites identified in celery leaves and petioles. Most phenolic compounds improved to some extent under salt stress after exogenous proline treatment (Figure 6D,E), consistent with Kusvuran et al. [27]. Proline (0.3 mM) increased benzoic acid, ferulic acid, caffeic acid, cinnamic acid, alizarin, and chlorogenic acid in celery leaves, and glutaric acid, ferulic acid, benzoic acid, and quercetin in petioles. Furthermore, NaCl (100 mM) increased kaempferol, 4-coumaric acid, quercetin, and rutin in celery leaves (Figure 6D), and 4-coumaric acid and gallic acid in petioles (Figure 6E). Notably, phenolic compounds have antioxidant activity [48]; therefore, it is necessary to gain insight into the correlation between polyphenolic compounds and the ROS-scavenging capacity of the ASA–GSH cycle. We found multiple or highly significant positive correlations between polyphenol levels and antioxidant activity in the celery leaves and petioles (Figure 11A,B). Additionally, highly positive correlations were found between total phenolic and chlorogenic acids in celery leaves (r = 1.00) and petioles (r = 0.96), respectively (Appendix A). Similarly, highly positive correlations were found between benzoic acid and SOD in celery leaves (r = 0.97) and petioles (r = 0.98), respectively (Appendix A). Therefore, we speculated that chlorogenic and benzoic acids play important antioxidant roles in celery resistance to salt stress.

ROS are considered to be toxic by products of aerobic metabolism [50]; they play a role in signaling mechanisms as second messengers [51]. Plants continuously produce ROS, such as singlet oxygen, O^2-^, H_2_O_2_, and OH-, during photosynthesis and respiration [52], which increase significantly in response to abiotic stresses [53]. Luo et al. [54] found that excessive accumulation of ROS under salt stress causes membrane damage. We found that the O^2-^ and H_2_O_2_ contents in celery leaves and petioles were significantly increased under salt stress compared with the control; however, exogenous proline decreased their content (Figure 7B,C). Furthermore, the tissue staining of celery leaves reflected H_2_O_2_ and O^2-^ levels (Figure 7A). Antioxidant enzymes and nonenzymatic antioxidant systems (ASA–GSH cycle) are the main forces for scavenging ROS and are essential for plant resilience [55]. The main enzymes that scavenge ROS in plants are SOD, CAT, and POD [56]. An increase in antioxidant enzyme activity was observed in rice plants exposed to salt [34]. Additionally, Parvaiz et al. [57] observed that SOD, CAT, APX, and GR activities increased under NaCl stress. We found that proline significantly increased SOD, CAT, and POD activities in celery under salt stress (Figure 8A–C). Therefore, we suggest that antioxidant enzymes protect celery from oxidative damage.

The ASA–GSH cycle mainly exists in the chloroplast, facilitating resistance to oxidative stress and scavenging active oxygen [58]. ASA and GSH regulate major cells and play a key role in antioxidant defense [59]. Furthermore, ASA is directly involved in scavenging ROS during plant growth and development [24]. Additionally, GSH maintains the integrity of cellular antioxidant machinery in response to stress [58]. We found that proline significantly increased ASA and GSH contents in celery leaves and petioles under salt stress (Figure 9A,B). GSH/GSSG showed a similar trend (Figure 9D); a higher GSH/GSSG reflected a higher antioxidant capacity. Therefore, these non-enzymatic antioxidants protected celery from oxidative damage. APX generates monohydroascorbic acid and uses ascorbic acid to reduce H_2_O_2_ to water [24]. GR plays a vital role in ROS detoxification and redox state maintenance in crops under environmental stress [60]. Furthermore, MDHAR limits the amount of MDHA radicals undergoing non-enzymatic disproportionation for DHA generation. Additionally, Kaya et al. [55] observed that salinity stress enhanced APX, GR, MDHAR, and DHAR activities. We found that celery leaves and petioles treated with proline under salt stress had significantly increased AAO, APX, GR, and DHAR activities (Figure 10A–C,E) and significantly reduced MDHAR activity (Figure 10D). This result suggests that exogenous proline protects celery from ROS-induced damage by promoting ASA and GSH regeneration, increasing the activity of ASA–GSH cycle enzymes, and activating the ASA–GSH cycle, thereby reducing oxidative damage and lipid peroxidation reactions. A similar effect of proline on antioxidant enzyme activity was found in other plants under different stresses [20,61].

## 4. Materials and Methods

### 4.1. Plant Materials and Growth Conditions

The test celery was “American Celery”, purchased from Jiayuguan Baoneng Agricultural Technology Co., Ltd. (Xincheng, Jiayuguan City, China), which has good disease resistance and high yield. Proline (≥99%) was purchased from Shanghai Macklin Biochemical Technology Company, Limited. The seedlings were planted in Lanzhou New District Modern Agricultural Investment Group Company Limited. After the seedlings grew to two leaves and one heart, vigorous seedlings with uniform growth were selected and transplanted into plastic pots (20 × 20 cm) containing cultivation substrate, grass charcoal, and vermiculite in a volume ratio of 3:1:1, respectively, with three plants per pot. The experiment was conducted in a glasshouse at Gansu Agricultural University in Lanzhou, northwestern China (36°05′39.86″ N, 103°42′31.09″ E). The air temperature was 15–20 ± 2 °C, the photoperiod was 13–14 h, the light intensity was 800–1200 µmol m^−2^ s^−1^ and the relative humidity was 60–70%.

### 4.2. Treatments and Experimental Design

The experimental treatment was initiated when the seedlings grew to two leaves and one heart. The experiment was completely randomized with four treatments. Three biological replicates were set up for each treatment, representing 39 seedlings per replicate and 117 seedlings per treatment.

The treatment combinations were as follows: CK (0 mM proline + 0 mM NaCl), proline (0.3 mM proline + 0 mM NaCl), N (0 mM proline +100 mM NaCl), and PN (0.3 mM proline +100 mM NaCl). Seedlings with two leaves and one heart were transplanted into plastic pots (20 × 20 cm) containing cultivation substrate, grass charcoal, and vermiculite in a volume ratio of 3:1:1. After 7 days, the foliage of the celery seedlings was sprayed with 0.3 mM proline solution at 9 am every day for 7 days. After that, the plants were watered with 300 mL 100 mM NaCl solution per pot thrice every 3 days. All indicators were evaluated on the eighth day after treatment. The samples were wrapped in tin foil, frozen in liquid nitrogen, and stored in an ultra-low temperature refrigerator at −80 °C to determine relevant indicators.

### 4.3. Morphological Index Determination

Plant height, stem diameter, and fresh and dry weights of shoots and roots were determined on the eighth day after treatment. Plant height is the length from the bottom of the stem to the top of the growing point, and stem diameter was measured using Vernier calipers. Next, the plants were harvested, and fresh and dry weights were obtained.

### 4.4. Leaf stomatal, Photosynthetic Gas Exchange Parameter and Calvin Cycle Enzyme Activity

Celery leaves (avoiding the main leaf veins) were cut into small slices (5 mm × 5 mm), placed in 10 mL centrifuge tubes, and fully submerged in 4% glutaraldehyde fixative for 2 h at room temperature. Afterward, it was rinsed with phosphate buffer (0.1 M PBS, pH = 6.8) for 40 min at 10 min intervals. Next, the samples were sequentially rinsed with different concentrations of ethanol, which was subsequently removed ethanol from the samples with different concentrations of tert-butanol. After, the samples were dried and sprayed with a layer of metal powder, and the stomata were observed and photographed using a scanning electron microscope (SEM, Hitachi-S3400N) [62]. The photosynthetic gas exchange parameters, including Gs, Pn, Ci, and Tr of celery leaves, were measured using a Ciras-2 portable photosynthesizer (PP System Inc., Amesbury, MA, USA) from 9:00 am to 11:00 am on sunny days [63]. The activities of key enzymes in the Calvin cycle were measured using the ELISA Kit (Yaji Biotech, Shanghai, China).

### 4.5. Chlorophyll Content and Chlorophyll Fluorescence Parameters

The chlorophyll content was determined as described by Moustakas et al. [64]. Additionally, the chlorophyll fluorescence parameters of celery leaves were measured using the Maxi Imaging PAM chlorophyll fluorescence apparatus, as described by Dong et al. [65] with some modifications. Next, the initial fluorescence (Fo), maximum fluorescence (Fm), steady-state fluorescence (Fs), maximum fluorescence yield (Fm′), and initial fluorescence in light (Fo′) were measured sequentially. Lastly, the Fv/Fm, Y(II), qP, qN, and excitation pressure of PSII (1-qP) were calculated.
Fv/Fm =(Fm−Fo)/Fm
$$Y\left(\mathrm{II}\right)=\left(\left({\mathrm{Fm}}^{\prime }-\mathrm{Fs}\right)/{\mathrm{Fm}}^{\prime }\right)$$
$$\mathrm{qP}\;=\left({\mathrm{Fm}}^{\prime }-\mathrm{Fs}\right)/\left({\mathrm{Fm}}^{\prime }-{\mathrm{Fo}}^{\prime }\right)$$
$$\mathrm{qN}\;=\left(\mathrm{Fm}-{\mathrm{Fm}}^{\prime }\right)/\left(\mathrm{Fm}-{\mathrm{Fo}}^{\prime }\right)$$

### 4.6. Analysis of Biological Membrane Damage and Osmotic Substances

REC was determined as described by Wang et al. [66]. After washing the celery leaves with deionized water, 12 round pieces of celery leaves with a diameter of 0.5 cm were collected from each treatment, evacuated with a vacuum pump for 30 min, surged on a shaking table for 3 h, and equilibrated at room temperature for 2 h. Next, the initial conductivity (S_1_) was measured using a Ddsj-308f conductivity meter (Shanghai INESA Scientific Instrument Co., Ltd., Shanghai, China). Subsequently, the final conductivity (S_2_) was measured in a boiling water bath for 30 min and cooled to room temperature. The conductivity of deionized water (S_0_) was used as a blank. REC was calculated as follows: REC (%) = (S_1_ − S_0_)/(S_2_ − S_0_) × 100

The MDA content in celery leaves and petioles was determined using the thiobarbituric acid method, as described by Bu et al. [40]. Furthermore, the soluble sugar content in celery leaves and petioles was determined by anthrone colorimetry, according to the method described by Li et al. [41]. Lastly, the soluble protein content in celery leaves and petioles was measured according to the method of Wang et al. [49].

### 4.7. Analysis of Phenolic Compounds

Polyphenols were analyzed using an HPLC (Waters Corp.) equipped with a 1525 pump and 2998 photodiode array detector. The phenolic compounds were determined as described by Wang et al. [49], with minor modifications, and were detected at 240, 280, and 322 nm (Table 1). 

### 4.8. Histochemical Staining and Quantitative Determination of O^2−^ and H_2_O_2_


O^2−^ and H_2_O_2_ were visually detected using DAB (1 mg·mL^−1^) and the NBT (0.1% *w*/*v*) uptake method, respectively, as described by Tang et al. [67]. The representative phenotypes were photographed with EPSON expression 11000XL color image scanner (WinRHIZO Pro LA2400, Regent Instruments Inc., Quebec City, QC, Canada). Additionally, O^2−^ and H_2_O_2_ contents were determined according to the method described by Bu et al. [40]. 

### 4.9. Antioxidant Enzyme Activity Assays

Celery leaves and petioles (0.5 g samples) were powdered in liquid nitrogen and suspended in 5 mL phosphate buffer (50 mM, pH 7.8) containing 5 mM EDTA, 2 mM ASA, and 2% polyvinylpyrrolidone (PVP). Next, SOD activity was determined by inhibiting the photochemical reduction of NBT, as previously described [41]. Additionally, CAT activity was measured at 240 nm by measuring H_2_O_2_ degradation. Lastly, POD activity was assayed with minor modifications, 0.1 mL of the enzyme extract was mixed with 2.6 mL guaiacol (0.3% in 50 mM phosphate buffer, pH 6.5) and 0.3 mL of 0.6% H_2_O_2_, and the change in absorbance was measured at 470 nm for 2 min.

### 4.10. Measurement of Non-Enzymatic Antioxidant Contents

ASA content was measured using the 2,6-dichloroindophenol stain method [68]. GSH and GSSG contents were determined according to the method described by Kaur et al. [69].

### 4.11. Ascorbic Acid–Glutathione Cycle-Related Enzyme Activity

AAO, DHAR, and MDHAR were determined according to the method described by Jimenez et al. [70] with some modifications. Briefly, celery leaves and petioles (0.5 g samples) were powdered in liquid nitrogen and suspended in 5 mL of phosphate buffer Tris-HCl (0.3 M mannitol, 1 mM EDTA, 1% BSA, 0.05% cysteine, 2% PVP, pH 7.2). After centrifugation at 16,000× *g* for 20 min at 4 °C, the supernatant was used to determine AAO, DHAR, and MDHAR activities. AAO activity was defined as the oxidation of 1 μmol ASA per minute per milligram of total protein. Moreover, DHAR activity was defined as the reduction of 1 μmol DHA per minute per milligram of total protein for one enzyme activity unit (U total protein mg^−1^). Additionally, MDHAR activity was defined as the oxidation of 1 μmol MDHA per minute per milligram of total protein. APX and GR were determined according to the method described by Kim et al. [71], with some modifications. Briefly, powdered celery leaves and petioles (0.8 g samples) were suspended in 5 mL of phosphate buffer (50 mM, pH 7.8) containing 0.1 mM EDTA, 0.3% Triton X-100, and 4% PVP. After centrifugation at 16,000× *g* for 15 min at 2 °C, the supernatant was used to determine APX and GR activities. APX activity was measured by the change in OD290 with 3 mL 50 mM Hepes-KOH (pH 7.6), 1 mM H_2_O_2_, 0.5 mM ASA, and 0.05 mL enzyme solution. APX activity was defined as the oxidation of 1 µM ASA per minute per milligram of total protein. GR activity was defined as the reduction of 1 mM GSSG per minute per milligram of total protein.

### 4.12. Statistical Analysis

All experiments were performed in triplicate, and the results are expressed as the mean ± standard error. ANOVA was performed using SPSS (version 20.0; SPSS Institute Inc., Armonk, NY, USA) and Duncan’s multiple range test with a probability level of 0.05 for comparison. All data were compiled using OriginPro 2022 (OriginLab Institute Inc., Northampton, MA, USA).

## 5. Conclusions

We investigated the mitigating effect of proline on celery plant damage under salt stress. NaCl (100 mM) inhibited the growth of celery plants; however, proline (0.3 mM) promoted celery growth and phenolic compound accumulation. Furthermore, proline reduced cell membrane damage and enhanced photosynthesis by improving chlorophyll fluorescence and Calvin cycle enzyme activity. Additionally, proline enhanced antioxidant capacity by increasing antioxidant enzyme activity and activating the ASA–GSH cycle, protecting celery from ROS induced by salt stress damage, thereby reducing salt stress toxicity and ensuring the normal physiological activities of cells. 

## Figures and Tables

**Figure 1 plants-12-00928-f001:**
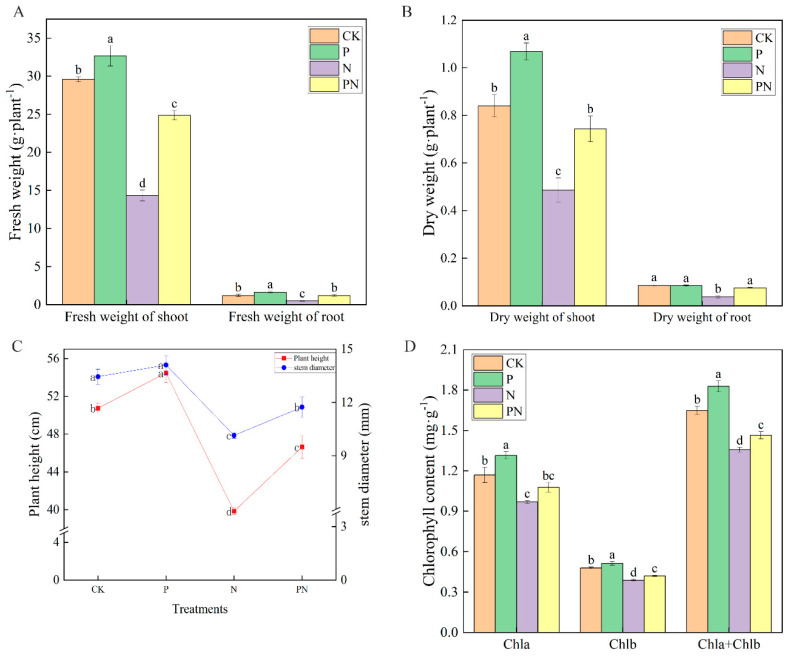
Effect of exogenous proline on celery’s growth and photosynthetic pigment content under salt stress. After planting the 35-day-old seedlings for 7 days, they were treated with 0.3 mM proline solution for 7 days, after which they were watered with 100 mM NaCl solution thrice every 3 days. (**A**) Fresh weight. (**B**) Dry weight. (**C**) Plant height and stem diameter. (**D**) Chlorophyll content. The results are expressed as the mean ± SE of three replicates, and the different letters denote the significant difference among treatments (*p* < 0.05), according to Duncan’s multiple tests. CK, control. P, 0.3 mM proline. N, 100 mM NaCl. PN, 100 mM NaCl+ 0.3 mM proline.

**Figure 2 plants-12-00928-f002:**
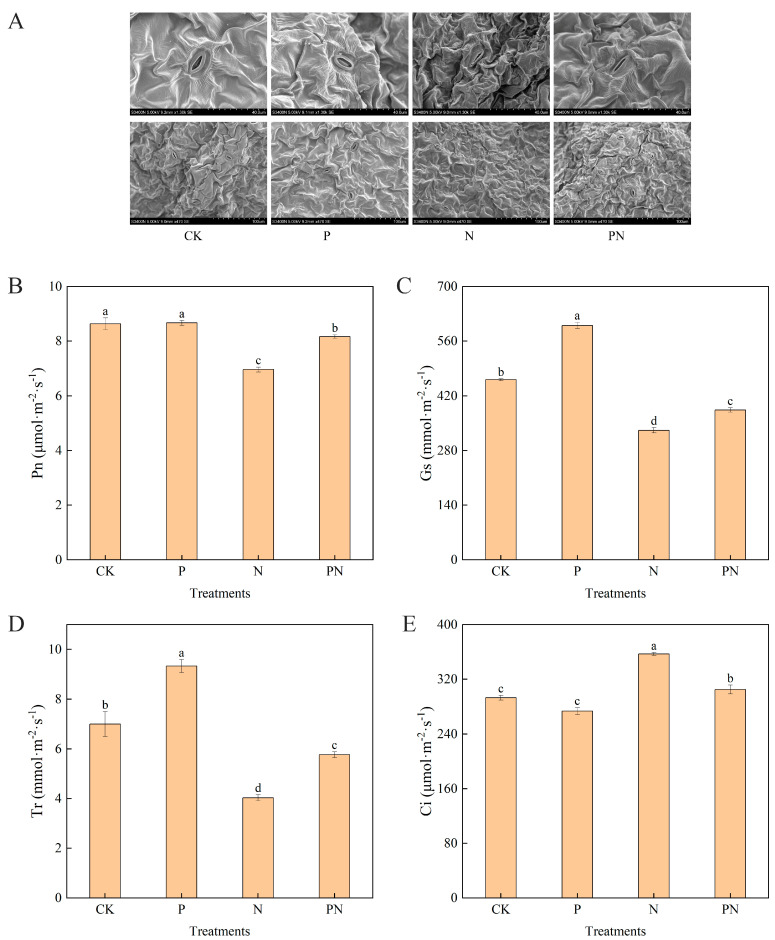
Effect of proline on stomatal and photosynthetic gas exchange parameters in celery under salt stress. After planting the 35-day-old seedlings for 7 days, they were treated with 0.3 mM proline solution for 7 days, after which they were watered with 100 mM NaCl solution thrice every 3 days. (**A**) Leaf stomatal morphology, 1300×, scale bars = 40 µm; 470×, scale bars = 100 µm. (**B**) Net photosynthetic rate (Pn). (**C**) Stomatal conductance (Gs). (**D**) Transpiration rate (Tr). (**E**) Intercellular CO_2_ concentration (C_i_). The results are expressed as the mean ± SE of three replicates, and the different letters denote the significant difference among treatments (*p* < 0.05), according to Duncan’s multiple tests. CK, control. P, 0.3 mM proline. N, 100 mM NaCl. PN, 100 mM NaCl+ 0.3 mM proline.

**Figure 3 plants-12-00928-f003:**
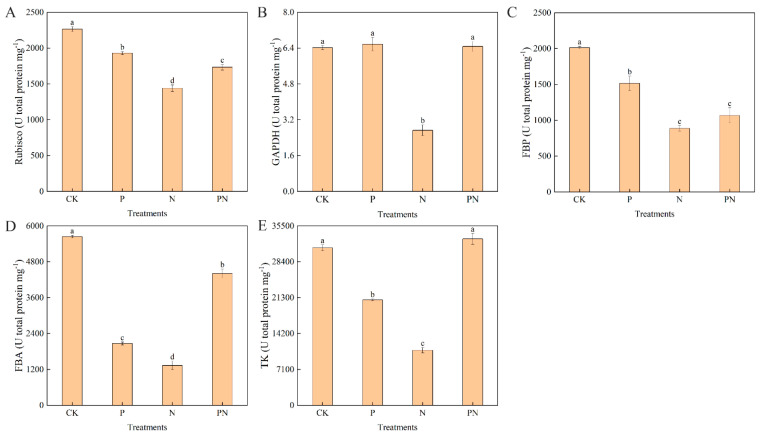
Effect of exogenous proline on the activity of key Calvin cycle enzymes in celery leaves under salt stress. After planting the 35-day-old seedlings for 7 days, they were treated with 0.3 mM proline solution for 7 days, after which they were watered with 100 mM NaCl solution thrice every 3 days. (**A**) 1,5-diphosphate carboxylase, Rubisco. (**B**) 3-Glyceraldehyde-phosphate dehydrogenase, GAPDH. (**C**) Fructose-1,6-bisphosphatase, FBPase. (**D**) Fructose 1,6-diphosphate aldolase, FBA. (**E**) Transketolase, TK. The results are expressed as the mean ± SE of three replicates, and the different letters denote the significant difference among treatments (*p* < 0.05), according to Duncan’s multiple tests. CK, control. P, 0.3 mM proline. N, 100 mM NaCl. PN, 100 mM NaCl+ 0.3 mM proline.

**Figure 4 plants-12-00928-f004:**
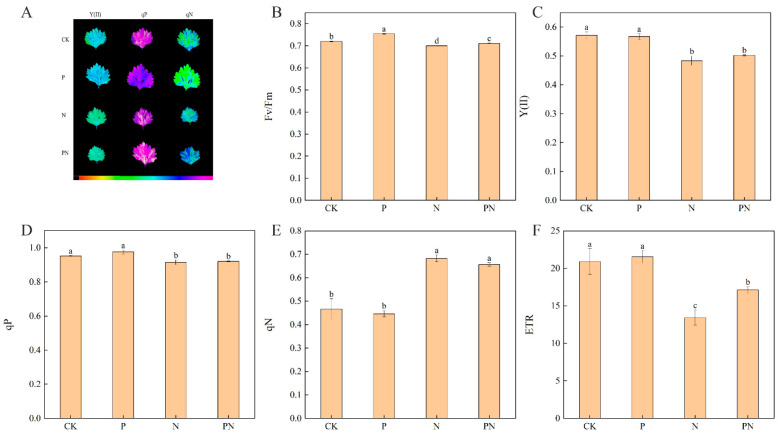
Exogenous proline effects on celery leaves’ chlorophyll fluorescence parameters under salt stress. After planting the 35-day-old seedlings for 7 days, they were treated with 0.3 mM proline solution for 7 days, after which they were watered with 100 mM NaCl solution thrice every 3 days. (**A**) Images of Y(II), qP and qN. (**B**) Maximum efficiency of PSII photochemistry, (Fv/Fm). (**C**) The actual photochemical efficiency of PSII, [Y(II)]. (**D**) Coefficient of photochemical quenching, (qP). (**E**) Coefficient of non-photochemical quenching, (qN). (**F**) Relative electron transfer rate, (ETR). The results are expressed as the mean ± SE of three replicates, and the different letters denote the significant difference among treatments (*p* < 0.05), according to Duncan’s multiple tests. CK, control. P, 0.3 mM proline. N, 100 mM NaCl. PN, 100 mM NaCl+ 0.3 mM proline.

**Figure 5 plants-12-00928-f005:**
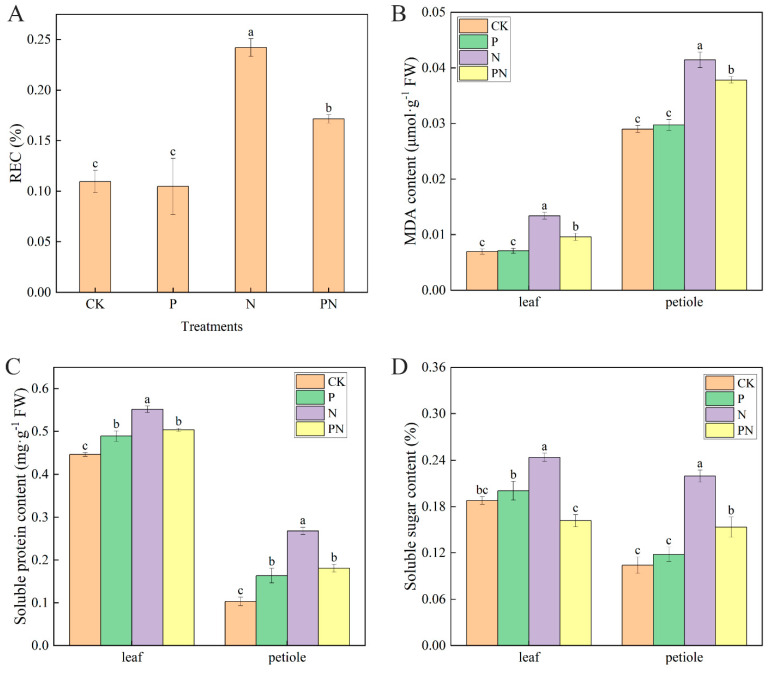
Effect of proline on cell membrane damage and osmotic substance content in celery under salt stress. After planting the 35-day-old seedlings for 7 days, they were treated with 0.3 mM proline solution for 7 days, after which they were watered with 100 mM NaCl solution thrice every 3 days. (**A**) Relative conductivity (REC). (**B**) Malondialdehyde (MDA) content. (**C**) Soluble protein content. (**D**) Soluble sugar content. The results are expressed as the mean ± SE of three replicates, and the different letters denote the significant difference among treatments (*p* < 0.05), according to Duncan’s multiple tests. CK, control. P, 0.3 mM proline. N, 100 mM NaCl. PN, 100 mM NaCl+ 0.3 mM proline.

**Figure 6 plants-12-00928-f006:**
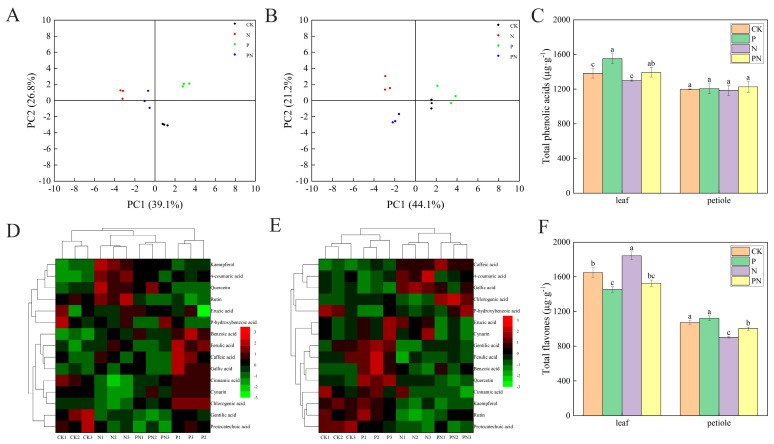
Statistical analysis of exogenous proline effect on bioactive metabolites in celery under salt stress. After planting the 35-day-old seedlings for 7 days, they were treated with 0.3 mM proline solution for 7 days, after which they were watered with 100 mM NaCl solution thrice every 3 days. (**A**) A plot of unsupervised principal component analysis (PCA) scores of phenolic metabolites in celery leaves. (**B**) PCA scores of phenolic metabolites in celery petioles. (**C**) Total phenolic acids. (**D**) Heat map of phenolic compound concentrations in celery leaves. (**E**) Heat map of phenolic compound concentrations in celery petioles. The colored areas respond to the concentration of different phenolic compounds in the group treated with exogenous proline under salt stress conditions, from low (green) to high (red). Each row represents one phenolic substance, and each column represents one treatment. (**F**) Total flavones. The results (Figure 6C,F) are expressed as the mean ± SE of three replicates, and the different letters denote the significant difference among treatments (*p* < 0.05), according to Duncan’s multiple tests. CK, control. P, 0.3 mM proline. N, 100 mM NaCl. PN, 100 mM NaCl+ 0.3 mM proline.

**Figure 7 plants-12-00928-f007:**
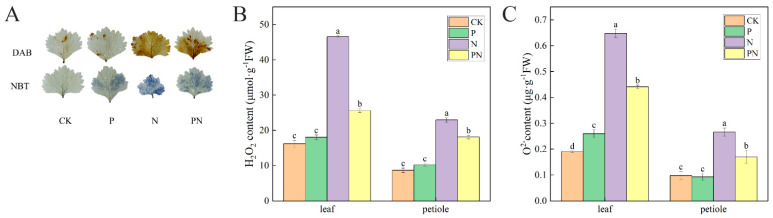
Effect of proline on O^2-^ and H_2_O_2_ accumulation in celery under salt stress. After planting the 35-day-old seedlings for 7 days, they were treated with 0.3 mM proline solution for 7 days, after which they were watered with 100 mM NaCl solution thrice every 3 days. (**A**) Diaminobenzidine (DAB) staining, nitro blue tetrazolium (NBT) staining. (**B**) H_2_O_2_ content. (**C**) O^2-^ content. CK, control. P, 0.3 mM proline. N, 100 mM NaCl. PN, 100 mM NaCl+ 0.3 mM proline. The results are expressed as the mean ± SE of three replicates, and the different letters denote the significant difference among treatments (*p* < 0.05), according to Duncan’s multiple tests. CK, control. P, 0.3 mM proline. N, 100 mM NaCl. PN, 100 mM NaCl+ 0.3 mM proline.

**Figure 8 plants-12-00928-f008:**
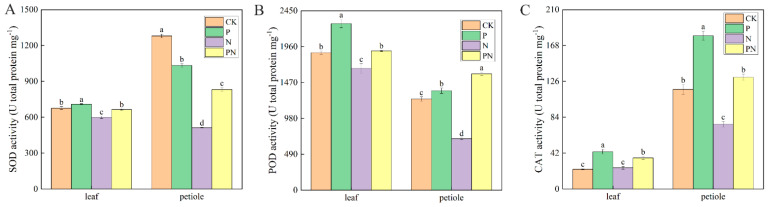
Antioxidant enzyme activity of celery under salt stress as affected by exogenous proline. After planting the 35-day-old seedlings for 7 days, they were treated with 0.3 mM proline solution for 7 days, after which they were watered with 100 mM NaCl solution thrice every 3 days. (**A**) SOD activity. (**B**) POD activity. (**C**) CAT activity. The results are expressed as the mean ± SE of three replicates, and the different letters denote the significant difference among treatments (*p* < 0.05), according to Duncan’s multiple tests. CK, control. P, 0.3 mM proline. N, 100 mM NaCl. PN, 100 mM NaCl+ 0.3 mM proline.

**Figure 9 plants-12-00928-f009:**
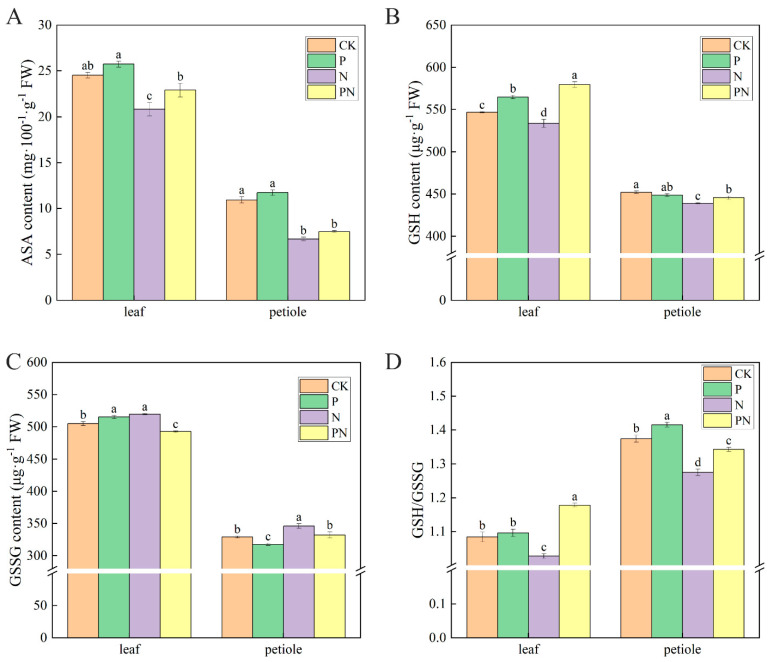
Effect of exogenous proline on the content of non-enzymatic antioxidant substances in celery under salt stress. After planting the 35-day-old seedlings for 7 days, they were treated with 0.3 mM proline solution for 7 days, after which they were watered with 100 mM NaCl solution thrice every 3 days. (**A**) Reduced ascorbic acid (ASA) content. (**B**) Glutathione (GSH) content. (**C**) Oxidized glutathione (GSSG) content. (**D**) GSH/GSSG. The results are expressed as the mean ± SE of three replicates, and the different letters denote the significant difference among treatments (*p* < 0.05), according to Duncan’s multiple tests. CK, control. P, 0.3 mM proline. N, 100 mM NaCl. PN, 100 mM NaCl+ 0.3 mM proline.

**Figure 10 plants-12-00928-f010:**
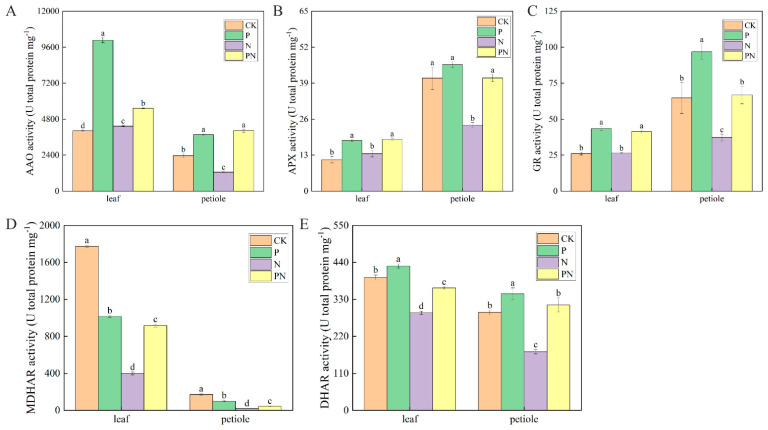
Effect of exogenous proline on the activity of ASA–GSH cycle-related enzymes in celery under salt stress. After planting the 35-day-old seedlings for 7 days, they were treated with 0.3 mM proline solution for 7 days, after which they were watered with 100 mM NaCl solution thrice every 3 days. (**A**) Ascorbate oxidase (AAO) activity. (**B**) Ascorbate peroxidase (APX) activity. (**C**) Glutathione reductase (GR) activity. (**D**) Monodehydroascorbate reductase (MDHAR) activity. (**E**) Dehydroascorbate reductase (DHAR) activity. The results are expressed as the mean ± SE of three replicates, and the different letters denote the significant difference among treatments (*p* < 0.05), according to Duncan’s multiple tests. CK, control. P, 0.3 mM proline. N, 100 mM NaCl. PN, 100 mM NaCl+ 0.3 mM proline.

**Figure 11 plants-12-00928-f011:**
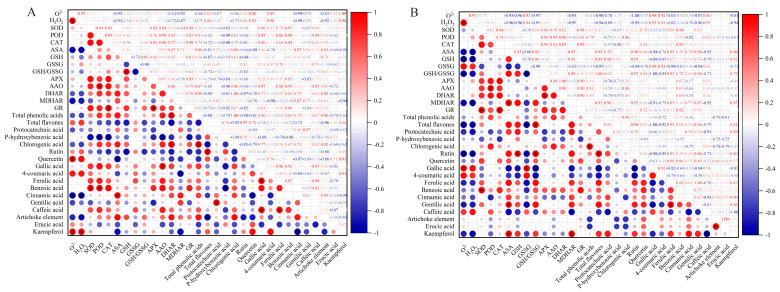
Pearson correlation analysis of exogenous proline effect on phenolic compounds and ascorbic acid (ASA)-glutathione (GSH) cycle in celery under salt stress. * represents significant correlations at *p <* 0.05. After planting the 35-day-old seedlings for 7 days, they were treated with 0.3 mM proline solution for 7 days, after which they were watered with 100 mM NaCl solution thrice every 3 days. (**A**) Celery leaves. (**B**) Celery petioles.

**Table 1 plants-12-00928-t001:** Phenolic compounds were determined by HPLC at three wavelengths.

Wavelength	240 nm	280 nm	322 nm
Phenolic compounds	Protocatechuic acid	4-Coumaric acid	Caffeic acid
	*P*-hydroxybenzoic acid	Cinnamic acid	Erucic acid
	Chlorogenic acid	Benzoic acid	Kaempferol
	Quercetin	Gallic acid	Cynarin
	Rutin	Ferulic acid	Gentilic acid

## Data Availability

All data, tables and figures in this manuscript are original.

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
