# Peer review of "Exogenous Proline Enhances Systemic Defense against Salt Stress in Celery by Regulating Photosystem, Phenolic Compounds, and Antioxidant System"

_plants, 2023, doi:10.3390/plants12040928_

Round 1
Reviewer 1 Report
I have reviewed the research article entitled "Exogenous Proline Enhances Systemic Defense Against Salt Stress in Celery by Regulating Photosystem, Phenolic Compounds and Antioxidant System", which has been submitted to Plants. This manuscript primarily focuses on evaluate the effects of proline application on the damage of celery plants under salt stress.
It can be seen that the trial was conducted under rigorous management.
The experimental design seems to be adequate.
I have only minor comments. For instance, please indicate that represent the 39 seedlings per replicate, since it has three biological replicates. 39 plants represent a clone. This is not clear.
In the material and methods section, explain why the proline application was set to 0.3 mM. Why proline application was not set in different concentrations?
I think the conclusions of the study are adequate and understandable.
Author Response
Response to Reviewer 1 Comments
Comments and Suggestions for Authors
I have reviewed the research article entitled “Exogenous Proline Enhances Systemic Defense Against Salt Stress in Celery by Regulating Photosystem, Phenolic Compounds and Antioxidant System”, which has been submitted to Plants. This manuscript primarily focuses on evaluate the effects of proline application on the damage of celery plants under salt stress. It can be seen that the trial was conducted under rigorous management. The experimental design seems to be adequate.
Response : Thank you for your encouraging remarks and valuable comments on our manuscript entitled “Exogenous Proline Enhances Systemic Defense Against Salt Stress in Celery by Regulating Photosystem, Phenolic Compounds and Antioxidant System” (ID: plants-2181227). These comments are all valuable and very helpful for revising and improving our paper. We have revised the manuscript substantially and hope it will be approved by you. We revised our manuscript using the “Track Changes” function in Microsoft Word and the revised portions were marked in red color.
Point 1: I have only minor comments. For instance, please indicate that represent the 39 seedlings per replicate, since it has three biological replicates. 39 plants represent a clone. This is not clear.
Response 1: Thank you very much. We have carefully considered the suggestion of Reviewer about the sentence, now it reads as:
“Three biological replicates were set up for each treatment, representing 39 seedlings per replicate and 117 seedlings per treatment”. (Lines 520-521)
Point 2: In the material and methods section, explain why the proline application was set to 0.3 mM. Why proline application was not set in different concentrations?
Response 2: Thank you for your kind comment, in the pre-experiment, we investigated the alleviating effect of different concentrations of proline (0 mM, 0.1 mM, 0.3 mM, 0.9 mM, 1.5 mM, 3 mM, 4.5 mM, 6 mM) on NaCl (100 mM) stress, and we found that Pro (0.3 mM) could significantly improve the growth of celery under salt stress and enhance photosynthesis. The related results have been published in Acta Agriculturae Universitatis Jiangxiensis , so in this experiment, we set the proline concentration to 0.3 mM.
Point 3: I think the conclusions of the study are adequate and understandable.
Response 3: Thank you for your valuable suggestion.

Reviewer 2 Report
Authors must give Latin name of celery and point the aim of the study which differ from predecessors. From conclusion it needs to delete the phrase about necessity of Future studies.
Author Response
Response to Reviewer 2 Comments
Thank you for your and valuable comments on our manuscript entitled “Exogenous Proline Enhances Systemic Defense Against Salt Stress in Celery by Regulating Photosystem, Phenolic Compounds and Antioxidant System” (ID: plants-2181227). These suggestions were valuable and very helpful in revising and improving our paper. We have carefully revised the manuscript and hope it will be approved by you. We revised our manuscript using the “Track Changes” function in Microsoft Word and the revised portions were marked in red color.
Point 1: Authors must give Latin name of celery and point the aim of the study which differ from predecessors.
Response 1: Thank you for your valuable suggestion, we have added the Latin name of celery (Apium graveliens L.). (Line 73)
We have pointed out and added where the purpose of the study differences from its predecessors, now it reads as:
“Although the effects of proline on plant salt stress have attracted widespread attention, most studies have focused on seed germination, seedling growth, and gas exchange. However, few studies have reported the physiological responses of celery to exogenous proline under salt stress. In this study, we investigated the effects of proline application on the growth and gas exchange parameters, stomatal morphology, Calvin cycle, chlorophyll fluorescence, membrane damage, phenolic compounds, and antioxidant system of celery under salt stress using the “American Celery.” Additionally, we elucidated the correlation between polyphenolic compounds and the ROS scavenging capacity of the ASA-GSH cycle. We employed exhaustive methods of studying stress to provide a theoretical and technological basis for cultivating high-quality and high-yield celery under saline conditions”. (Lines 83-94)
Point 2: From conclusion it needs to delete the phrase about necessity of Future studies.
Response 2: Thank you for your kind comment, we have deleted the phrase about necessity of Future studies. (Lines 635-638)

Reviewer 3 Report
Manuscript entitled “Exogenous Proline Enhances Systemic Defense Against Salt Stress in Celery by Regulating Photosystem, Phenolic Compounds and Antioxidant System” by Gao et al describes a comparison study of the effects of foliar spraying of proline on celery plants that are later exposed to saline stress. The response of plants to stress is addressed through the study of different parameters, both morphological (height, biomass) and physiological (photosynthesis-related parameters, secondary metabolites, antioxidant systems). The authors carry out an experiment comparing plants fumigated or not with proline, which are exposed to control conditions or saline stress (exposure to NaCl).
Proline pretreatment as a system to improve the effects of saline stress in plants has already been widely described in the literature. Nonetheless, the main novelty in this work lies in the approach to the study of the effects of stress, which is quite exhaustive and checks the plant response to stress observing a great variety of parameters, as well as the study system, the crop celery. From my point of view, although the work seems correct in terms of experimental design and data collection and some of the results are interesting, it presents some issues that need to be revised throughout the manuscript to consider this work suitable for publication.
Main issues:
1) English needs a considerable revision. Some examples are enumerated in my additional comments below, but I think that the manuscript would be greatly benefited by a review by a native speaker English.
2) I find that the descriptions of the results are often exposed in a difficult way for a future reader. I recommend the authors to try to improve this description, my advice here is that for each parameter studied the comparison between treatments try to describe:
· What has been the result of treatment with proline in the absence of stress (P vs CK)
· What has been the result of salt stress (N vs Ck).
· the presence of proline affects the result obtained in saline stress? (PN vs N or also PN vs CK)
3) Enzyme activities:
To ensure a proper comparison between samples, enzyme activities should be expressed in specific activity units: enzyme units/ total protein quantity.
a) Figure 3: expressing your activities referred to ml of extract could lead to activity differences that may be due to different extraction efficiency (samples with less protein extracted will have less activity for any enzyme).
b) Figure 10: enzyme activities should not be referred to fresh weight. The variations observed in your samples could be due to cell water content variations…
For this reason, I kindly urge to the authors to revise their enzyme activity data, make the calculations, and present the corresponding figures in specific activity units. I am sure that the results obtained from some comparisons between treatments will be more confident (and probably different from those showed in the present version)
4) The text has numerous deficiencies when citing references, particularly the use of “et al.” without italics or the use of author name and surname, or full capitalized surnames. I have pointed out some examples in the additional comments below, but please check ALL the manuscript. Also, the list of references contains a lot of “[J]”, the names of the journals appear both with or without italics, and in extended or abbreviate format…please check the required format of the references in the author’s instructions of the journal and make the appropriate corrections.
Additional comments:
Regarding Abstract
Lines 12-13: Please change “The results showed that 0.3 mM proline exhibited significantly plant biomass” with “The results showed that proline-treated plants exhibited a significant increase in plant biomass”
Line 16: Please delete “were supported”
Lines 17-18: please change “Proline (0.3 mM) significantly suppressed the increase in relative conductivity and malondialdehyde content, reducing biological membrane damage.” With “Also, Proline spraying significantly suppressed the increase in relative conductivity and malondialdehyde content caused by salt stress, suggesting a reduction in biological membrane damage.”
Lines 18-19: the sentence is confusing…”foliar spraying of proline resulted in the accumulation of these parameters restored” and after that “their visualization was reduced accordingly”, please rewrite
Regarding Introduction
Line 35: please change “7%” with “7 %”, please do the same with the rest of percentage values of the manuscript
Line 40 please change “oxygen species (ROS)” with “reactive oxygen species (ROS)”
Line 49: please change “glutamate pathway[12].” With “glutamate pathway [12].”
Lines 50-53: please change “Numerous researches have shown that proline plays an essential role in response to abiotic stresses, such as salinity, high temperature, drought and other adversities can cause the accumulation of proline in the plant [13, 14]. A positive correlation between proline accumulation and plant tolerance to various abiotic stresses [15].” With “Numerous research evidence has shown that proline plays an essential role in response to abiotic stress conditions, such as salinity, high temperature, drought, and others, where has been observed an accumulation of proline in the plant when exposed to these adverse conditions [13, 14]. Also, it has been described a positive correlation between proline accumulation and plant tolerance to various abiotic stresses [15].”
Lines 55 and below: “et al.” should appear in italics. Please correct it in the rest of the times that it appears in the manuscript
Line 55: please change “foliar spraying of 20 mM of proline” with “foliar spraying with a 20 mM proline solution”
Line 56: “Brassica juncea” should appear in italics. Also, MESSEDI D et al. [18] should appear in lowercase (Messedi)
Line 57:” Cakile marítima” should appear in italics
Line 60: “SHAHID M A et al.” should appear in lowercase
Line 61: “Pisum sativum” should appear in italics
Line 63: BORGO L et al. should appear in lowercase
Line 63: plsae change “proline can contributes” with “proline can contribute”
Line 67: “showed showed”, please correct the duplication
Line 68: please correct “O. sativa” with “Oryza sativa”, also it should appear in italics
Line 69: SOD, POD and CAT abbreviatures should be defined
Line 73: Umbelliferae should appear in italics
Regarding results
Line 89: CK abbreviature should be defined
Line 90: Again, P abbreviature should be defined
Line 91: Again, N abbreviature should be defined
Line 91: Again, PN abbreviature should be defined
Line 103-105: Details about plant age and duration of the treatment should be included in all figure legends
Lines 106-117: Pn, Gs, Tr, Ci abbreviations should be defined the first time that appear in the main text
Lines 119-123: the statistical meaning of the letters in the figures should be detailed in each figure legend, it is not enough mentioning ““same as below” in figure legend 1. Please correct this in the rest of figures of the manuscript
Line 124: Rubisco, GAPDH, FBPase, FBA and TK abbreviatures should be defined
Lines 137-143: Fv/Fm, Y(II), qP, qN and ETR abbreviations should be defined
Line 148: Eelative or Relative?
Line 150: I doubt that the term “biofilm” could be applied ito any part of celery plant of cells….”membrane damage”?
Lines 151-157: REC,MDA abbreviations should be defined the first time that appear in the main text
Line 158: figure 5C please change (mg g-1) with (mg g-1 FW)
Line 160: REC abbreviation should appear in figure legend
Line 171: what molecule is “Artichoke element”? Also it not appears in the heatmaps 6D or 6E
Line 178 : please change (µg g-1) with (µg g-1 FW) in Figures 6C and 6F. Also, resolution of figures 6D ane &3 should be improved, the compound names are barely visible
Line 189: NBT and DAB abbreviations should be defined the first time that appear in the main text
Line 210: ASA, GSH and GSSG) abbreviations should be defined the first time that appear in the main text
Line 217: figure 9 please change (mg g-1) with (mg g-1 FW)
Line 224: AAO, APX, GR, MDHAR and DHAR abbreviations should be defined the first time that appear in the main text
Line 231: I am very surprised that minutes -1 appear in figure 10….please check your definitions of unit of enzymatic activity used in the assays
Line 249: figure 11. Resolution of the panels 11A and 11B should be improved. Also, the correlation data (r and P) should be included as a supplementary table
Line 261: please change “plants[29].” With “plants [29].”
Line 263: Why an increase of Na content in leaves causes a decrease in chlorophyll content?.
Why salt stress causes an “expansion of chrloplast membranes”. How is this connected to a decrease in chlorophyll content?
Line 284: please check the names of the stages of the calvin cycle….there is not a “reduction of phosphate” stage
Line 301: pleaee change “ZHONG M et al” with “Zhong et al.”
Line 313: please change “A Nakhaie et al” with “Nakhaie et al.” and also “D Messedi et al.” with “Messedi et al.”
Line 313: C. maritima should appear with its complete name and in italics
Lines 340-341: please change “Chong Xie et al [46]” with “Xie et al. [46]”
Line 342: please change “Kusvuran et al. [27] with “Kusvuran et al. [27]”
Lines 346.354: This paragraph should be in the results section. Only describes again the results of figure 6A and 6B.
Line 377: ROS abbreviation has already been defined in line 40
Line 378: “they play a role in signaling mechanisms as second messengers”
Regarding Materials and Methods
Line 419: The source of the “American celery” line should be referenced or described
Line 491: please correct “measureed”
Line 501: please correct “repre sentative”
Line 522: please replace “cycteine” with “cysteine”
Line 531: please correct “tritonX-100” with “triton X-100”
Regarding conclusion:
Line 546: again, I do not think that the term “biofilm” could be applied to celery plants o celery cells.
Lines 546-548: proline concentration is repeated three times in the paragraph…I do not think that this is necessary.
Author Response
Response to Reviewer 3 Comments
Comments and Suggestions for Authors
Manuscript entitled “Exogenous Proline Enhances Systemic Defense Against Salt Stress in Celery by Regulating Photosystem, Phenolic Compounds and Antioxidant System” by Gao et al describes a comparison study of the effects of foliar spraying of proline on celery plants that are later exposed to saline stress. The response of plants to stress is addressed through the study of different parameters, both morphological (height, biomass) and physiological (photosynthesis-related parameters, secondary metabolites, antioxidant systems). The authors carry out an experiment comparing plants fumigated or not with proline, which are exposed to control conditions or saline stress (exposure to NaCl).
Proline pretreatment as a system to improve the effects of saline stress in plants has already been widely described in the literature. Nonetheless, the main novelty in this work lies in the approach to the study of the effects of stress, which is quite exhaustive and checks the plant response to stress observing a great variety of parameters, as well as the study system, the crop celery. From my point of view, although the work seems correct in terms of experimental design and data collection and some of the results are interesting, it presents some issues that need to be revised throughout the manuscript to consider this work suitable for publication.
Response : Thank you for your valuable comments on our manuscript entitled “Exogenous Proline Enhances Systemic Defense Against Salt Stress in Celery by Regulating Photosystem, Phenolic Compounds and Antioxidant System.” (ID: plants-2181227). These comments are all valuable and very helpful for revising and improving our paper, as well as the important guiding significance to our researches. We have revised the manuscript substantially and hope it will be approved. We revised our manuscript using the “Track Changes” function in Microsoft Word and the revised portions were marked in red color.
Our point-by-point responses to yours comments are detailed below, in which the line numbers are marked according to the completed revised manuscript:
Point 1: English needs a considerable revision. Some examples are enumerated in my additional comments below, but I think that the manuscript would be greatly benefited by a review by a native speaker English.
Response 1: Thank you for your valuable suggestion. We have reviewed the manuscript with the help of a native English speaker, carefully checked the grammar and structure of the entire text. The following is a certification of English editing services:
Point 2: I find that the descriptions of the results are often exposed in a difficult way for a future reader. I recommend the authors to try to improve this description, my advice here is that for each parameter studied the comparison between treatments try to describe: What has been the result of treatment with proline in the absence of stress (P vs CK), What has been the result of salt stress (N vs Ck), the presence of proline affects the result obtained in saline stress? (PN vs N or also PN vs CK).
Response 2: Thank you very much! We have carefully considered the question. We have made efforts to improve the description of the results, from Figure 1 to Figure 10, for each parameter studied, we describe in detail the comparison between treatments, systematically describing and adding P vs CK, N vs Ck, PN vs N and PN vs CK. (Lines 102-104, 133-134, 148-153, 175-176, 190-195, 200-201, 215-222, 247-250, 267-270, 309-311)
Point 3: Enzyme activities: To ensure a proper comparison between samples, enzyme activities should be expressed in specific activity units: enzyme units/ total protein quantity.
- a) Figure 3: expressing your activities referred to ml of extract could lead to activity differences that may be due to different extraction efficiency (samples with less protein extracted will have less activity for any enzyme).
- b) Figure 10: enzyme activities should not be referred to fresh weight. The variations observed in your samples could be due to cell water content variations…
For this reason, I kindly urge to the authors to revise their enzyme activity data, make the calculations, and present the corresponding figures in specific activity units. I am sure that the results obtained from some comparisons between treatments will be more confident (and probably different from those showed in the present version)
Response 3: Thank you very much. We have carefully considered the suggestions of Reviewer about enzyme activities units, and we have changed the enzyme activity units.
In Figure 3, we carefully checked the enzyme activity data according to the suggestions of Reviewer and changed the enzyme activity unit to U/(total protein quantityl). (Lines 155-156)
In Figure 10, we have carefully corrected the enzyme activity data according to the suggestions of Reviewer, and we have changed the enzyme activity units, and new findings were obtained from the comparison between treatments. (Lines 313-314)
Point 4: The text has numerous deficiencies when citing references, particularly the use of “et al.” without italics or the use of author name and surname, or full capitalized surnames. I have pointed out some examples in the additional comments below, but please check ALL the manuscript. Also, the list of references contains a lot of “[J]”, the names of the journals appear both with or without italics, and in extended or abbreviate format…please check the required format of the references in the author’s instructions of the journal and make the appropriate corrections.
Response 4: Thank you for your valuable suggestion. We have carefully checked all references in the manuscript according to the reference format required in the journal's author's instructions, and “et al.” has all been italicized, the authors' first and last names have all been standardized, “[J]” has all been removed from the reference list, and the journal names have all been changed to extended format and italics.
Point 5: Lines 12-13: Please change “The results showed that 0.3 mM proline exhibited significantly plant biomass” with “The results showed that proline-treated plants exhibited a significant increase in plant biomass”
Response 5: Thank you for your valuable suggestion. We have changed “The results showed that 0.3 mM proline exhibited significantly plant biomass” with “The results showed that proline-treated plants exhibited a significant increase in plant biomass”. (Lines 13-14)
Point 6: Line 16: Please delete “were supported”
Response 6: Thank you very much! We have deleted “were supported”. (Line 17)
Point 7: Lines 17-18: please change “Proline (0.3 mM) significantly suppressed the increase in relative conductivity and malondialdehyde content, reducing biological membrane damage.” With “Also, Proline spraying significantly suppressed the increase in relative conductivity and malondialdehyde content caused by salt stress, suggesting a reduction in biological membrane damage.”
Response 7: We have changed “Proline (0.3 mM) significantly suppressed the increase in relative conductivity and malondialdehyde content, reducing biological membrane damage.” With “Also, Proline spraying significantly suppressed the increase in relative conductivity and malondialdehyde content caused by salt stress, suggesting a reduction in biological membrane damage.” (Lines 19-21)
Point 8: Lines 18-19: the sentence is confusing…”foliar spraying of proline resulted in the accumulation of these parameters restored” and after that “their visualization was reduced accordingly”, please rewrite
Response 8: We have rewritten “foliar spraying of proline resulted in the accumulation of these parameters restored” and after that “their visualization was reduced accordingly,” now it reads as: “Moreover, salt stress resulted in hydrogen peroxide, superoxide anions and 4-coumaric acid accumulation in celery, and their contents were reduced after foliar spraying of proline.” (Lines 21-25)
Regarding Introduction
Point 9: Line 35: please change “7%” with “7 %”, please do the same with the rest of percentage values of the manuscript
Response 9: Thank you for your valuable suggestion. We changed “7%” to “7 %” and completed the same process for the other percentage values in the manuscript. (Line 38)
Point 10: Line 40 please change “oxygen species (ROS)” with “reactive oxygen species (ROS)”
Response 10: We have changed “oxygen species (ROS)” with “reactive oxygen species (ROS)”. (Line 40)
Point 11: Line 49: please change “glutamate pathway[12].” With “glutamate pathway [12].”
Response 11: We have changed “glutamate pathway[12].” With “glutamate pathway [12].” (Line 50)
Point 12: Lines 50-53: please change “Numerous researches have shown that proline plays an essential role in response to abiotic stresses, such as salinity, high temperature, drought and other adversities can cause the accumulation of proline in the plant [13, 14]. A positive correlation between proline accumulation and plant tolerance to various abiotic stresses [15].” With “Numerous research evidence has shown that proline plays an essential role in response to abiotic stress conditions, such as salinity, high temperature, drought, and others, where has been observed an accumulation of proline in the plant when exposed to these adverse conditions [13, 14]. Also, it has been described a positive correlation between proline accumulation and plant tolerance to various abiotic stresses [15].”
Response 12: We have changed “Numerous researches have shown that proline plays an essential role in response to abiotic stresses, such as salinity, high temperature, drought and other adversities can cause the accumulation of proline in the plant [13, 14]. A positive correlation between proline accumulation and plant tolerance to various abiotic stresses [15].” With “Numerous research evidence has shown that proline plays an essential role in response to abiotic stress conditions, such as salinity, high temperature, drought, and others, where has been observed an accumulation of proline in the plant when exposed to these adverse conditions [13, 14]. Also, it has been described a positive correlation between proline accumulation and plant tolerance to various abiotic stresses [15].” (Lines 54-59)
Point 13: Lines 55 and below: “et al.” should appear in italics. Please correct it in the rest of the times that it appears in the manuscript
Response 13: We have changed “et al.” to italics (et al.) and corrected the rest of the manuscript where it appears.
Point 14: Line 55: please change “foliar spraying of 20 mM of proline” with “foliar spraying with a 20 mM proline solution”
Response 14: We have changed “foliar spraying of 20 mM of proline” with “foliar spraying with a 20 mM proline solution”. (Line 61)
Point 15: Line 56: “Brassica juncea” should appear in italics. Also, MESSEDI D et al. [18] should appear in lowercase (Messedi)
Response 15: We have changed “Brassica juncea” to italics (Brassica juncea), and MESSEDI D has been changed to lowercase (Messedi). (Line 62)
Point 16: Line 57:” Cakile marítima” should appear in italics
Response 16: We have changed “Cakile marítima” to italic (Cakile marítima). (Lines 62-63)
Point 17: Line 60: “SHAHID M A et al.” should appear in lowercase
Response 17: We have changed “SHAHID M A et al.” to “Shahid M A et al.” (Line 65)
Point 18: Line 61: “Pisum sativum” should appear in italics
Response 18: We have changed “Pisum sativum” to italic (Pisum sativum). (Line 66)
Point 19: Line 63: BORGO L et al. should appear in lowercase
Response 19: We have changed “BORGO L et al.” to “Borgo L et al.”. (Line 68)
Point 20: Line 63: plsae change “proline can contributes” with “proline can contribute”
Response 20: We have changed “proline can contributes” with “proline can contribute”. (Line 69)
Point 21: Line 67: “showed showed”, please correct the duplication
Response 21: We have deleted the duplicate content “showed”. (Line 73)
Point 22: Line 68: please correct “O. sativa” with “Oryza sativa”, also it should appear in italics
Response 22: We have changed “O. sativa” to “Oryza sativa” and shown it in italics (Oryza sativa). (Line 74)
Point 23: Line 69: SOD, POD and CAT abbreviatures should be defined
Response 23: We have defined the abbreviations for SOD (superoxide dismutase), POD (peroxidase) and CAT (catalase). (Line 75)
Point 24: Line 73: Umbelliferae should appear in italics
Response 24: We have changed “Umbelliferae” to italic (Umbelliferae). (Line 80)
Regarding results
Point 25: Line 89: CK abbreviature should be defined
Response 25: Thank you for your valuable suggestion. We have defined the abbreviations CK (Control). (Line 98)
Point 26: Line 90: Again, P abbreviature should be defined
Response 26: We have defined the abbreviations P (Proline). (Line 98)
Point 27: Line 91: Again, N abbreviature should be defined
Response 27: We have defined the abbreviations N (NaCl). (Line 99)
Point 28: Line 91: Again, PN abbreviature should be defined
Response 28: We have defined the abbreviations PN (Proline+NaCl). (Line 102)
Point 29: Line 103-105: Details about plant age and duration of the treatment should be included in all figure legends
Response 29: We have added details about plant age and duration of the treatment in all figure legends.
Point 30: Lines 106-117: Pn, Gs, Tr, Ci abbreviations should be defined the first time that appear in the main text
Response 30: We have defined the abbreviations Pn (Net photosynthetic rate), Gs(Stomatal conductance), Tr (Transpiration rate), Ci (Intercellular CO2 concentration). (Lines 125-129)
Point 31: Lines 119-123: the statistical meaning of the letters in the figures should be detailed in each figure legend, it is not enough mentioning “same as below” in figure legend 1. Please correct this in the rest of figures of the manuscript
Response 31: Thank you for your valuable suggestion. We have detailed the statistical significance of the letters in the graphs in each legend.
Point 32: Line 124: Rubisco, GAPDH, FBPase, FBA and TK abbreviatures should be defined
Response 32: We have defined the abbreviations Rubisco (1,5-diphosphate carboxylase), GAPDH (3-Glyceraldehyde-phosphate dehydrogenase), FBPase (Fructose-1,6-bisphosphatase), FBA (Fructose 1,6-diphosphate aldolase) and TK (Trans-ketolase). (Lines 146-148)
Point 33: Lines 137-143: Fv/Fm, Y(II), qP, qN and ETR abbreviations should be defined
Response 33: We have defined the abbreviations Fv/Fm (Maximum efficiency of PSII photochemistry), Y(II) (The actual photochemical efficiency of PSII), qP (Coefficient of photochemical quenching), qN (Coefficient of non-photochemical quenching) and ETR (Relative electron transfer rate). (Lines 168-170,173)
Point 34: Line 148: Eelative or Relative?
Response 34: Thank you for your valuable suggestion, it is “Relative”. (Line 183)
Point 35: Line 150: I doubt that the term “biofilm” could be applied ito any part of celery plant of cells….”membrane damage?
Response 35: We have changed the word “biofilm” with “cell membrane”.
Point 36: Lines 151-157: REC, MDA abbreviations should be defined the first time that appear in the main text
Response 36: We have defined the abbreviations REC (Relative conductivity) and MDA (Malondialdehyde). (Lines 188, 192)
Point 37: Line 158: figure 5C please change (mg g-1) with (mg g-1 FW)
Response 37: We have changed “(mg g-1)” with “(mg g-1 FW)” in figure 5C.
Point 38: Line 160: REC abbreviation should appear in figure legend
Response 38: We have added REC abbreviation in figure legend. (Line 206)
Point 39: Line 171: what molecule is “Artichoke element”? Also it not appears in the heatmaps 6D or 6E
Response 39: Thank you for your valuable suggestion, “Artichoke element” is “Cynarin”, it appears heatmaps 6D and 6E. This was an error caused by the translation at the time, and we have completed the changes in Table 1 and the corresponding location.
Point 40: Line 178 : please change (µg g-1) with (µg g-1 FW) in Figures 6C and 6F. Also, resolution of figures 6D ane &3 should be improved, the compound names are barely visible
Response 40: We have changed (µg g-1) to (µg g-1 FW) in Figures 6C and 6F, and improved the resolution of the figures 6D ane &3. (Lines 229-230)
Point 41: Line 189: NBT and DAB abbreviations should be defined the first time that appear in the main text
Response 41: We have defined the abbreviations NBT (Nitro blue tetrazolium) and DAB (Diaminobenzidine). (Line 243)
Point 42: Line 210: ASA, GSH and GSSG) abbreviations should be defined the first time that appear in the main text
Response 42: We have defined the abbreviations ASA (Reduced ascorbic acid), GSH (Glutathione), GSSG (Oxidized glutathione). (Lines 282-283)
Point 43: Line 217: figure 9 please change (mg g-1) with (mg g-1 FW)
Response 43: We have changed “(mg g-1)” with “mg g-1 FW” in Figure 9. (Lines 288-289)
Point 44: Line 224: AAO, APX, GR, MDHAR and DHAR abbreviations should be defined the first time that appear in the main text
Response 44: We have defined the abbreviations AAO (Ascorbate oxidase), APX (Ascorbate peroxidase), GR (Glutathione reductase), MDHAR (Monodehydroascorbate reductase) and DHAR (Dehydroascorbate reductase). (Lines 302-305)
Point 45: Line 231: I am very surprised that minutes-1 appear in figure 10….please check your definitions of unit of enzymatic activity used in the assays
Response 45: Thank you for your valuable suggestion. We carefully checked and corrected the enzyme activity units. (Lines 313-314)
Point 46: Line 249: figure 11. Resolution of the panels 11A and 11B should be improved. Also, the correlation data (r and P) should be included as a supplementary table
Response 46: Thank you for your valuable suggestion. We have improved the resolution of Figures 11A and 11B outside and have included relevant data (r and P) in the supplemental tables , due to the previous change in enzyme activity units, we have also modified Figures 11A and 11B accordingly. (Lines 337-338)
Point 47: Line 261: please change “plants[29].” With “plants [29].”
Response 47: We have changed “plants[29].” with “plants [29].” (Line 352)
Point 48: Line 263: Why an increase of Na content in leaves causes a decrease in chlorophyll content?.
Why salt stress causes an “expansion of chrloplast membranes”. How is this connected to a decrease in chlorophyll content?
Response 48: Thank you for your valuable suggestion. Because higher Na+ levels enhance the activity of enzymes involved in the degradation of chlorophyll and thus promote chlorophyll degradation, thus we hypothesized that elevated Na+ levels would lead to a decrease in chlorophyll content in plant leaves.
Thank you very much. We have changed “which may be due to excess Na+ in celery leaves or expansion of chloroplast membranes [30].” with “which may be due to excess Na+ in celery leaves [30]”. (Line 354)
Point 49: Line 284: please check the names of the stages of the calvin cycle….there is not a “reduction of phosphate” stage
Response 49: We have changed “reduction of phosphate” with “reduction of 3-phosphoglyceric acid” . (Lines 374-375)
Point 50: Line 301: pleaee change “ZHONG M et al” with “Zhong et al.”
Response 50: We have changed “ZHONG M et al” with “Zhong et al.” (Line 391)
Point 51: Line 313: please change “A Nakhaie et al” with “Nakhaie et al.” and also “D Messedi et al.” with “Messedi et al.”
Response 51: We have changed “A Nakhaie et al” with “Nakhaie et al.” and also “D Messedi et al.” with “Messedi et al.” (Line 402-403)
Point 52:Line 313: C. maritima should appear with its complete name and in italics
Response 52: We have added the full name of maritima, “Cakile maritima”, in italics (Cakile maritima). (Line 404)
Point 53: Lines 340-341: please change “Chong Xie et al [46]” with “Xie et al. [46]”
Response 53: We have changed “ Chong Xie et al [46]” with “Xie et al. [46]”. (Line 429)
Point 54: Line 342: please change “Kusvuran et al. [27] with “Kusvuran et al. [27]”
Response 54: We have changed “ Kusvuran et al. [27]” with “Kusvuran et al. [27]”. (Line 431)
Point 55: Lines 346.354: This paragraph should be in the results section. Only describes again the results of figure 6A and 6B.
Response 55: Thank you for your valuable suggestion. We have deleted this paragraph. (Lines 436-443)
Point 56: Line 377: ROS abbreviation has already been defined in line 40
Response 56: We have deleted the definition of ROS. (Line 465)
Point 57: Line 378: “they play a role in signaling mechanisms as second messengers”
Response 57: We have changed “ it plays a role in receptor signaling as a second messenger” with “they play a role in signaling mechanisms as second messengers”. (Lines 466-467)
Regarding Materials and Methods
Point 58: Line 419: The source of the “American celery” line should be referenced or described
Response 58: Thank you for your valuable suggestion. The test celery was "American Celery," purchased from Jiayuguan Baoneng Agricultural Technology Co., Ltd.( Xincheng, Jiayuguan City, China) , which has good disease resistance and high yield. (Lines 505-506)
Point 59: Line 491: please correct “measureed”
Response 59: We have changed “measureed” with “measured”. (Line 576)
Point 60: Line 501: please correct “repre sentative”
Response 60: We have changed “repre sentative” with “representative”. (Line 586)
Point 61: Line 522: please replace “cycteine” with “cysteine”
Response 61:We have changed “cycteine” with “ cysteine”. (Line 606)
Point 62: Line 531: please correct “tritonX-100” with “triton X-100”
Response 62: We have changed “tritonX-100” with “triton X-100”. (Line 615)
Regarding conclusion:
Point 63: Line 546: again, I do not think that the term “biofilm” could be applied to celery plants o celery cells.
Response 63: Thank you for your valuable suggestion. We have changed the word “biofilm” with “cell membrane”. (Line 630)
Point 64: Lines 546-548: proline concentration is repeated three times in the paragraph…I do not think that this is necessary.
Response 64: Thank you very much. We have deleted the last two proline concentrations and kept only the first proline concentration. (Lines 629, 631)

Round 2
Reviewer 3 Report
The authors have addressed some of the issues previously pointed out and the revised version of the manuscript has improved with the changes made. Still, there a couple of important issues that should be addressed before considering the acceptance of the work.
Main issues:
a) Figures 3 and 10: total protein quantity units (mg or µg) should be specified, so: Y axis legends should appear as (U total protein mg-1) or (U total protein µg-1)
b) Figure 6 : description of the statistical analysis of figures 6C, 6E still absent in figure 6 legend.
c) Figure 8: SOD, POD and CAT activities still remain expressed in units different from specific activity units. Please make the appropriate calculations, represent them accordingly and if necessary, change the description/discussion of the results.
d) Lines 1274-1290: Please check your definition of AAO, DHAR, MDHAR, APX and GR specific activities.
Additional comments:
Regarding Abstract
Line 56 : please correct “an-alyzed”
Line 60: please correct “fluo-rescence”
Line 69: please correct “gluta-thione”
Line 70: please correct “acti-vating”
Line 157: please remove “ before “Numerous”
Line 162: please remove “ after “[15].”
Line 165: as I mentioned in my previous review, the correct format of citing should be 1st author surname et al., without name initials….”Messedi et al, “ and not “Messedi D et al”
Please correct accordingly in lines 168, 171,170, 178-179….
Regarding Results:
Lines 328, 348, 409,464,572,591,637,690: please correct “de-note”
Line 423: Please correct “re-spectively”
Line 479: please correct “cynarinand” with “cynarin and“
Line 702: please correct “peti-oles”
Line 762: please correct” plants [29].,”
Line 764: please correct “Na+” with “Na+” (+ in superscript)
Line 766: please correct “growth[31].”
Line 842: Please correct “un-der”
Line 843: Please correct ”Rubico”
Line 853: please correct “as-similation”
Line 855: As a suggestion, perhaps sounds better the sentence “This result is consistent with the observations made by Zhong et al [31].” Also, I think that the results of these authors should be briefly described (to justify the consistency of the results of the present manuscript)
Line 866 “et al.” should appear in italics
Line 854: please correct “con-tribute”
Line 972: please correct “in-sight”
Line 973: please correct “ca-pacity”
Line 975: Please correct “Ad-ditionally”
Line 988: please correct “de-creased”
Line 993: Please correct “ex-posed”
Line 1008: please correct “detoxify-cation”
Line 1068: please correct “sig-nificantly”
Line 1282: please correct “oxida-tion”
Regarding supplementary data:
Supplementary table legend should be included in an appropriate section of the manuscript, and should also be cited in the main text when necessary. Also, I do not think that the English correction archives should be included in the final version.
Author Response
Comments and Suggestions for Authors
The authors have addressed some of the issues previously pointed out and the revised version of the manuscript has improved with the changes made. Still, there a couple of important issues that should be addressed before considering the acceptance of the work.
Response : Thank you for your encouraging remarks and valuable comments again on our manuscript entitled “Exogenous Proline Enhances Systemic Defense Against Salt Stress in Celery by Regulating Photosystem, Phenolic Compounds and Antioxidant System.” (ID: plants-2181227). These comments were very helpful for us to revise and improve the paper. We have carefully considered the reviewers' comments and revised the manuscript carefully in the hope that it will be approved. We revised our manuscript using the “Track Changes” function in Microsoft Word and the revised portions were marked in red color.
Our point-by-point responses to yours comments are detailed below, in which the line numbers are marked according to the completed revised manuscript:
Main issues:
Point 1: a) Figures 3 and 10: total protein quantity units (mg or µg) should be specified, so: Y axis legends should appear as (U total protein mg-1) or (U total protein µg-1)
Response 1: Thank you for your valuable suggestion. We have changed the total protein quantity units in Figure 3 and Figure 10, and Y axis legends are now shown as (U total protein mg-1). (Lines 155, 315)
Point 2: b) Figure 6 : description of the statistical analysis of figures 6C, 6E still absent in figure 6 legend.
Response 2: Thank you very much! We have added the description of the statistical analysis of figures 6C, 6F in Figure 6 legend. (Lines 240, 242)
Point 3: c) Figure 8: SOD, POD and CAT activities still remain expressed in units different from specific activity units. Please make the appropriate calculations, represent them accordingly and if necessary, change the description/discussion of the results.
Response 3: Thank you for your valuable suggestion. We have recalculated the SOD, POD and CAT activity and changed the enzyme activity units, which are now expressed in units (U total protein mg-1). Meanwhile, we have also modified the description of the results, now it reads as: “Figures 8A, B, and C show celery leaves and petioles' SOD, POD, and CAT activities under a salt environment, respectively. Compared with CK, P treatment significantly increased the SOD, POD, and CAT activities, excluding the SOD activity in the petioles. The SOD, POD, and CAT activities in the PN group showed similar trends. Compared with CK, N treatment significantly decreased the SOD, POD, and CAT activities, excluding the CAT activity in the leaf.Compared with N, the SOD, POD and CAT activities were increased in the leaves and petiole of the PN group.” (Lines 264-272)
Point 4: d) Lines 1274-1290: Please check your definition of AAO, DHAR, MDHAR, APX and GR specific activities.
Response 4: We have carefully checked and modified the definitions of AAO, DHAR, MDHAR, APX, and GR specific activities, now they read as: “AAO activity was defined as the oxidation of 1 μmol ASA per minute per milligram of total protein. DHAR activity was defined as the reduction of 1 μmol DHA per minute per milligram of total protein for one enzyme activity unit (U total protein mg-1). MDHAR activity was defined as the oxidation of 1 μmol MDHA per minute per milligram of total protein. APX activity was defined as the oxidation of 1 µM ASA per minute per milligram of total protein. GR activity was defined as the reduction of 1 mM GSSG per minute per milligram of total protein.” (Lines 610-624)
Additional comments:
Regarding Abstract
Point 5: Line 56 : please correct “an-alyzed”
Response 5: Thank you for your valuable suggestion. We have changed “an-alyzed” with “analyzed”. (Line 11)
Point 6: Line 60: please correct “fluo-rescence”
Response 6: We have changed “fluo-rescence” with “fluorescence”. (Line 15)
Point 7: Line 69: please correct “gluta-thione”
Response 7: We have changed “gluta-thione” with “glutathione”. (Line 28)
Point 8: Line 70: please correct “acti-vating”
Response 8: We have changed “acti-vating” with “activating”. (Line 29)
Point 9: Line 157: please remove “ before “Numerous”
Response 9: We have removed “ before “Numerous”. (Line 54)
Point 10: Line 162: please remove “ after “[15].”
Response 10: We have removed“after “[15].” (Line 59)
Point 11: Line 165: as I mentioned in my previous review, the correct format of citing should be 1st author surname et al., without name initials….”Messedi et al, “ and not “Messedi D et al”
Response 11: Thank you for your valuable suggestion. We have changed “Messedi D et al.” with “Messedi et al.”. (Line 62)
Point 12: Please correct accordingly in lines 168, 171,170, 178-179….
Response 12: We have changed “Shahid M A et al.” with “Shahid et al.”. (Line 65)
We have changed “AliNakhaie et al.” with “Nakhaie et al.”. (Line 67), “Borgo L et al.” with “Borgo et al.”. (Line 68), “Marcin Naliwajski et al.” with “Naliwajski et al.”. (Line 76), “Messedi et al.” with “Messedi et al.”. (Line 408), “Kusvuran et al.” with “Kusvuran et al.”. (Line 408), “Kusvuran et al.” with “Kusvuran et al.”. (Line 456), “Xiaofeng Luo et al.” with “Luo et al.”. (Line 475), “Parvaiz Ahmad et al.” with “Parvaiz et al.”. (Line 484), “Ruifang Bu et al.” with “Bu et al.”. (Line 580), “Cheng Wang et al.” with “Wang et al.”. (Line 583), “Chaonan Tang et al.” with “Tang et al.”. (Line 592).
Regarding Results:
Point 13: Lines 328, 348, 409,464,572,591,637,690: please correct “de-note”
Response 13: We have changed “de-note” with “denote”. (Lines 142, 163, 184, 208, 259, 278, 298, 323)
Point 14: Line 423: Please correct “re-spectively”
Response 14: We have changed “re-spectively” with “respectively”. (Line 199)
Point 15: Line 479: please correct “cynarinand” with “cynarin and“
Response 15: We have changed “cynarinand” with “cynarin and”. (Lines 223-224)
Point 16: Line 702: please correct “peti-oles”
Response 16: We have changed “peti-oles” with “petioles”. (Lines 266)
Point 17: Line 762: please correct” plants [29].,”
Response 17: We have changed “plants [29].,” with “plants [29].” (Lines 354)
Point 18: Line 764: please correct “Na+” with “Na+” (+ in superscript)
Response 18: We have changed “Na+” with “Na+”. (Lines 356)
Point 19: Line 766: please correct “growth[31].”
Response 19: We have changed “growth[31].” with “growth [31]”. (Line 356)
Point 20: Line 842: Please correct “un-der”
Response 20: We have changed “un-der” with “under”. (Line 380)
Point 21: Line 843: Please correct ”Rubico”
Response 21: We have changed “Rubico” with “Rubisco”. (Line 380)
Point 22: Line 853: please correct “as-similation”
Response 22: We have changed “as-similation” with “assimilation”. (Line 391)
Point 23: Line 855: As a suggestion, perhaps sounds better the sentence “This result is consistent with the observations made by Zhong et al [31].” Also, I think that the results of these authors should be briefly described (to justify the consistency of the results of the present manuscript)
Response 23: Thank you for your valuable suggestion. We have added the results of these authors, now it reads as: “This result is consistent with that of Zhong et al. [37], they found TGase positively regulated photosynthesis by activating the Calvin cycle enzymes and inducing changes in cellular redox homeostasis in tomato.” (Lines 392-395)
Point 24: Line 866 “et al.” should appear in italics
Response 24: We have changed “et al.” with “et al.”. (Line 407)
Point 25: Line 854: please correct “con-tribute”
Response 25: We have changed “con-tribute” with “contribute”. (Line 392)
Point 26: Line 972: please correct “in-sight”
Response 26: We have changed “in-sight” with “insight”. (Line 460)
Point 27: Line 973: please correct “ca-pacity”
Response 27: We have changed “ca-pacity” with “capacity”. (Line 461)
Point 28: Line 975: Please correct “Ad-ditionally”
Response 28: We have changed “Ad-ditionally” with “Additionally”. (Line 463)
Point 29: Line 988: please correct “de-creased”
Response 29: We have changed “de-creased” with “decreased”. (Line 477)
Point 30: Line 993: Please correct “ex-posed”
Response 30: We have changed “ex-posed” with “exposed”. (Line 482)
Point 31: Line 1008: please correct “detoxify-cation”
Response 31: We have changed “detoxify-cation” with “detoxifycation”. (Line 496)
Point 32: Line 1068: please correct “sig-nificantly”
Response 32: We have changed “sig-nificantly” with “significantly”. (Line 501)
Point 33: Line 1282: please correct “oxida-tion”
Response 33: We have changed “oxida-tion” with “oxidation”. (Line 617)
Regarding supplementary data:
Point 34: Supplementary table legend should be included in an appropriate section of the manuscript, and should also be cited in the main text when necessary. Also, I do not think that the English correction archives should be included in the final version.
Response 34: Thank you for your valuable suggestion. We have added supplementary table legend in the appropriate section of the manuscript, and they are cited in the main text. (Lines 330, 332, 336, 339, 340, 466, 468)
We have deleted the English correction archives in the final version.

Round 3
Reviewer 3 Report
Just a couple minor issues regarding supplementary material that remain to be addressed in the final version of the manuscript:
a) A supplementary material section description remains to be added (perhaps between conclusions and author contributions, check the journal template). In this section, Supplementary Tables 1 and 2 legends should be included.
b) The excel archive incudes a supplementary table 1 and, but when referenced in the main text of the manuscript, sometimes appear the reference to a “Supplementary Table 1.2” (lines 708, 710, 714,, 992. 994) . What is table 1.2? because I do not see any indication inside the “Supplementary table 1 “ tab in the excel file. Please check it and include an appropriate description inside table 1 and 2 tabs if there is any table 1.1, 1.2 or 2.1 2.2. Also include them in the supplementary table legends already mentioned in a).
Author Response
Response to Reviewer 3 Comments
Comments and Suggestions for Authors
Just a couple minor issues regarding supplementary material that remain to be addressed in the final version of the manuscript:
Response : Thank you for your valuable suggestion again on our manuscript entitled “Exogenous Proline Enhances Systemic Defense Against Salt Stress in Celery by Regulating Photosystem, Phenolic Compounds and Antioxidant System.” (ID: plants-2181227). We have revised the manuscript carefully in the hope that it will be approved. We revised our manuscript using the “Track Changes” function in Microsoft Word and the revised portions were marked in red color.
Our point-by-point responses to yours comments are detailed below, in which the line numbers are marked according to the completed revised manuscript:
Point 1: a) A supplementary material section description remains to be added (perhaps between conclusions and author contributions, check the journal template). In this section, Supplementary Tables 1 and 2 legends should be included.
Response 1: Thank you very much! We have added a supplementary material section description between the conclusions and the author contributions, now it reads as: “Supplementary Table 1 provides Pearson's correlation analysis of phenolic compounds (Protocatechuic acid, P-hydroxybenzoic acid, Chlorogenic acid, Rutin, Quercetin, Gallic acid, 4-coumaric acid, Ferulic acid, Benzoic acid, Cinnamic acid, Gentilic acid, Caffeic acid, Cynarin, Erucic acid, Kaempferol, Total phenolic acids, Total flavones) and ASA-GSH cycle (O2-, H2O2, SOD, POD, CAT, ASA, GSH, GSSG, GSH/GSSG, APX, AAO, DHAR, MDHAR, GR) of celery leaves, which includes the correlation data. * denotes correlation coefficients that are significant at p<0.05 level. Supplementary Table 2 provides Pearson's correlation analysis of phenolic compounds (Protocatechuic acid, P-hydroxybenzoic acid, Chlorogenic acid, Rutin, Quercetin, Gallic acid, 4-coumaric acid, Ferulic acid, Benzoic acid, Cinnamic acid, Gentilic acid, Caffeic acid, Cynarin, Erucic acid, Kaempferol, Total phenolic acids, Total flavones) and ASA-GSH cycle (O2-, H2O2, SOD, POD, CAT, ASA, GSH, GSSG, GSH/GSSG, APX, AAO, DHAR, MDHAR, GR) of celery petioles, which includes the correlation data. * denotes correlation coefficients that are significant at p<0.05 level.” (Lines 651-663)
Point 2: b) The excel archive incudes a supplementary table 1 and, but when referenced in the main text of the manuscript, sometimes appear the reference to a “Supplementary Table 1.2” (lines 708, 710, 714,, 992. 994) . What is table 1.2? because I do not see any indication inside the “Supplementary table 1 “ tab in the excel file. Please check it and include an appropriate description inside table 1 and 2 tabs if there is any table 1.1, 1.2 or 2.1 2.2. Also include them in the supplementary table legends already mentioned in a).
Response 2: Thank you for your valuable suggestion. We have no table 1.1, 1.2 or 2.1 2.2, and we have changed “Supplementary Table 1.2” with “Supplementary Tables 1 and 2”. (Lines 330, 332, 336, 467, 469)
